# Small molecules as potent biphasic modulators of protein liquid-liquid phase separation

W. Michael Babinchak [1], Benjamin K. Dumm[1], Sarah Venus[2], Solomiia Boyko[1], Andrea A. Putnam[2,3], Eckhard Jankowsky[2] & Witold K. Surewicz [1]✉

Liquid-liquid phase separation (LLPS) of proteins that leads to formation of membrane-less organelles is critical to many biochemical processes in the cell. However, dysregulated LLPS can also facilitate aberrant phase transitions and lead to protein aggregation and disease. Accordingly, there is great interest in identifying small molecules that modulate LLPS. Here, we demonstrate that 4,4'-dianilino-1,1'-binaphthyl-5,5'-disulfonic acid (bis-ANS) and similar compounds are potent biphasic modulators of protein LLPS. Depending on context, bis-ANS can both induce LLPS de novo as well as prevent formation of homotypic liquid droplets. Our study also reveals the mechanisms by which bis-ANS and related compounds modulate LLPS and identify key chemical features of small molecules required for this activity. These findings may provide a foundation for the rational design of small molecule modulators of LLPS with therapeutic value.

[1] Department of Physiology and Biophysics, Case Western Reserve University, Cleveland, OH, USA. [2] Center for RNA Science & Therapeutics, Case Western Reserve University, Cleveland, OH, USA. [3]Present address: Department of Molecular Biology and Genetics, Johns Hopkins School of Medicine, Baltimore, MD, USA. ✉email: wks3@case.edu

L iquid−liquid phase separation (LLPS) mediates the formation of many intracellular membrane-less organelles (MLOs), including the nucleolus, nuclear paraspeckles, stress granules, and many other ribonucleoprotein granules[1–3]. This diverse functionality over a large number of MLO subtypes reflects the capacity for LLPS to finely tune biochemical reactions through rapid assembly and disassembly of organelles as well as inclusion or exclusion of specific chemical components[1,4].

Many proteins associated with MLOs form liquid-like droplets in the test tube[5–9], which has provided a platform for investigating the physical forces underlying LLPS. Weak, multivalent homotypic protein−protein interactions are critical for the formation and dynamic nature of many phase-separated droplets[1,3,10–12]. Additionally, heterotypic charge-based interactions are important in many phase-separating systems involving RNAs[13–15]. In polypeptide-RNA systems that undergo LLPS, low RNA concentrations can promote LLPS through constructive electrostatic (ionic charge−charge) interactions with protein[14,16,17]. However, at higher RNA concentrations, decondensation (or a reentrant phase transition back into a single phase[14,16]) results from protein charge overscreening and, hence, a long-range electrostatic repulsion[14]. Such reentrant phase transitions are hypothesized to be a pervasive, general occurrence in many multivalent systems involving heterotypic interactions[18]. In contrast to classical complex coacervates in which the condensation and decondensation regimes reflect the number of charged residues within and the charge density of each polyelectrolyte, the decondensation regime for polypeptide-RNA systems is also regulated by additional droplet-stabilizing, short-range cation−π interactions that become more significant at higher RNA to protein ratios[17,19,20].

Despite the recent finding that RNAs can tune protein LLPS behavior[21], a fundamental understanding of how LLPS per se is regulated by the cell remains relatively elusive and uncovering these details has become more urgent since the finding that dysregulation of LLPS is associated with disease[22,23]. In particular, prolonged LLPS of proteins such as fused in sarcoma (FUS)[24], the heterogeneous nuclear ribonucleoprotein A1[5], Huntingtin[25], and the transactive response DNA binding protein of 43 kDa (TDP-43)[26] can directly promote protein aggregation into amyloid fibrils, a hallmark of many neurodegenerative diseases[27]. As a result, there is rapidly emerging interest in identifying small molecules that could regulate MLOs. Efforts in this regard have focused on molecules that disrupt LLPS[28,29], though a chemical chaperone has been shown to promote LLPS at very high (0.5−2 M) concentrations[30]. However, there is growing evidence that MLOs can also be beneficial to normal functioning of the cell[4,31,32]. In fact, identification of small molecule *modulators* of LLPS—compounds that can disrupt aberrant phase transitions that form aggregates while preserving functionally important LLPS—might be even more critical to identifying new therapies. The chemical nature of such molecules might engender the capacity to both facilitate condensation de novo and also prevent excessive droplet formation, thereby modulating LLPS in a contextual manner.

In this study, we demonstrate that the small molecule, 4,4′-dianilino-1,1′-binaphthyl-5,5′-disulfonic acid (bis-ANS, 1), and similar compounds can act as potent modulators of LLPS of the TDP-43 low complexity domain (LCD). Remarkably, this modulation is biphasic: at low concentrations these molecules strongly promote TDP-43 LLPS, while at higher concentrations they disrupt liquid droplets through a reentrant phase transition driven by electrostatic repulsion. Even though a reentrant phase transition has been described for peptide/protein-RNA systems[14,19], our studies indicate that bis-ANS and related small molecules induce LLPS by a fundamentally different mechanism. We also identify the specific chemical features that are crucial to modulation of phase behavior by

small molecules and demonstrate that this modulatory capacity extends to a number of proteins. The present findings not only demonstrate the capacity of small molecules to regulate LLPS but may also provide a foundation for identifying and optimizing compounds that could modulate protein LLPS for therapeutic benefit.

## Results

**Bis-ANS modulates the liquid−liquid phase separation of TDP-43 LCD.** Bis-ANS (1) is a fluorescent molecule that has been used to probe exposed hydrophobic patches in proteins and to monitor the formation of protein aggregates[33,34]. The compound is a bivalent naphthalene sulfonate with aniline moieties (Fig. 1a). We initially sought to use this dye to monitor oligomerization and fibrillization of the TDP-43 LCD, a domain that drives pathological aggregation and LLPS[6,26]. However, during the course of these studies we made an unexpected observation that bis-ANS has a profound effect on the ability of TDP-43 LCD to undergo LLPS.

Phase separation of the TDP-43 LCD has been most extensively studied at pH around 6[6,26,35,36]. At higher pH, the protein has a much greater propensity not only to undergo LLPS but also to aggregate[26], which complicates biophysical studies of LLPS. Turbidity measurements used to assess the extent of LLPS and a phase diagram based on those measurements are shown in Supplementary Fig. 1a, b. These data clearly demonstrate that, at pH 6 in the absence of salt, the TDP-43 LCD remains in a single phase and that LLPS under these conditions is observed only above the saturation concentration ($c_{sat}$) of ~35 μM (Supplementary Fig. 1c, d). Consistent with previous data[6,35,36], addition of NaCl promotes homotypic interactions that induce formation of liquid droplets. However, this effect is only observed at relatively high protein concentrations or very high salt concentrations: the $c_{sat}$ for homotypic LLPS in the presence of 0.15, 0.5, and 1 M NaCl is 14.1, 9.5, and 4.1 μM, respectively (Supplementary Fig. 1c−e).

Remarkably, under conditions that do not induce homotypic LLPS (20 mM phosphate, pH 6, no salt), the addition of as little as 5 μM bis-ANS to the solution of TDP-43 LCD resulted in the formation of liquid droplets, even at physiologically relevant TDP-43 concentrations (5 μM)[37] (Fig. 1b). Bis-ANS appeared to be actively participating within droplets, as evidenced by the fluorescence images in Fig. 1b that rely on blue-green emission of bis-ANS to visualize droplets. To gain more detailed insight, we monitored solution turbidity as a function of bis-ANS concentration in the presence of multiple protein concentrations (Fig. 1c, in which representative turbidity traces are depicted as a function of the bis-ANS to TDP-43 LCD molar ratio). Lower ratios of bis-ANS to protein (<2:1) resulted in a robust increase in turbidity, indicating the formation of condensates that was confirmed by fluorescence microscopy (Fig. 1b, top panel). At a ratio of ~2:1, a well-defined maximum was observed and was followed by a gradual decline in turbidity at higher ratios. Eventually, no turbidity was detectable, indicating a second (or reentrant) phase transition back into the bulk phase. Complete dissolution of droplets under these conditions was verified by microscopy (Fig. 1b, bottom panel). A turbidity-based phase diagram relating bis-ANS and protein concentration is shown in Fig. 1d, demonstrating that such a biphasic behavior was observed in experiments at many protein concentrations, even as low as 2.5 μM and extending up to 15 μM.

To verify that droplets indeed contain both bis-ANS and TDP-43 LCD, we performed colocalization studies using fluorescence microscopy with Alexa Fluor 594-labeled TDP-43 LCD (AF594-LCD). These studies revealed that, as expected, both species

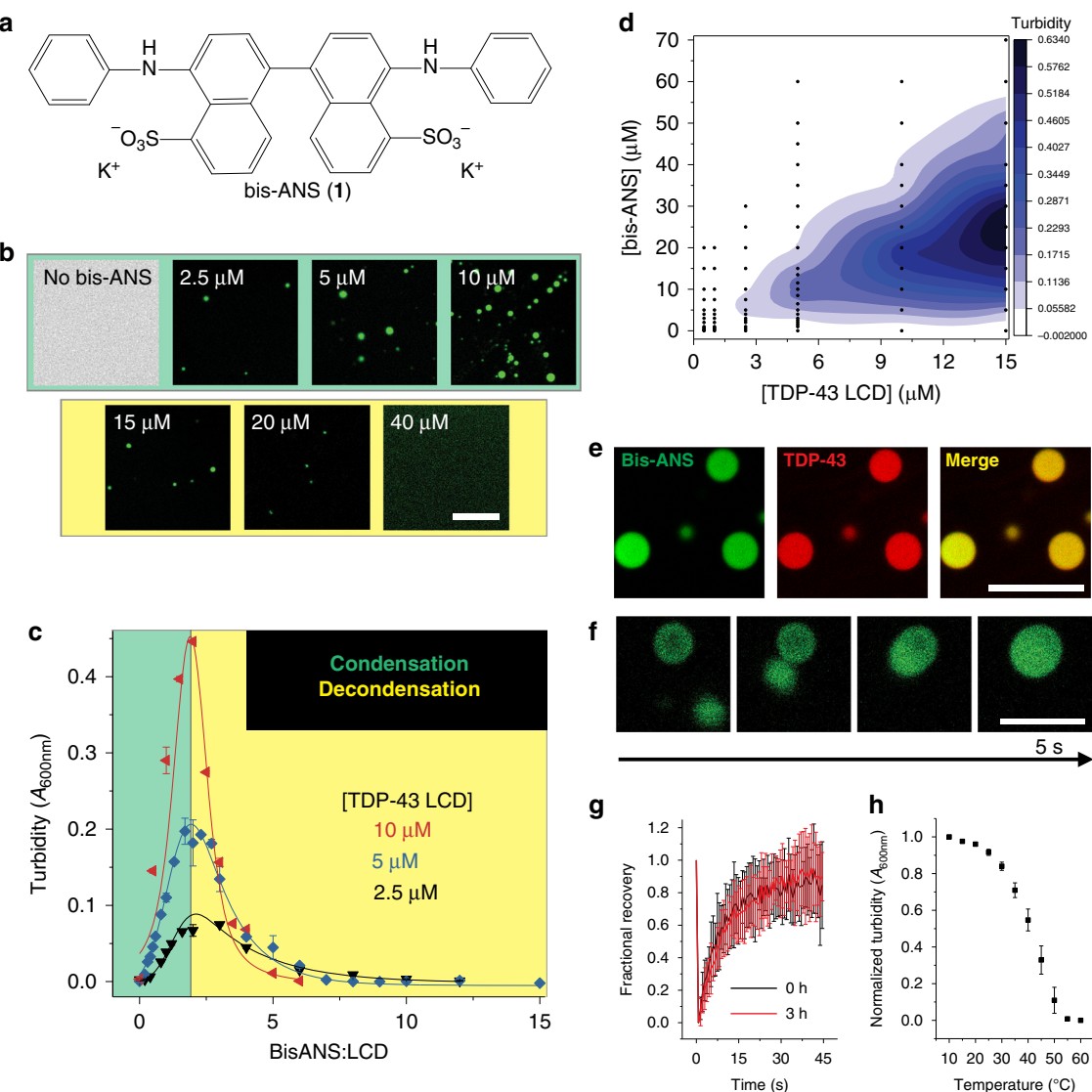

**Fig. 1 Bis-ANS induces LLPS of the TDP-43 LCD. a** Chemical structure of 4,4′-dianilino-1,1′-binaphthyl-5,5′-disulfonic acid (bis-ANS, **1**). **b** Representative fluorescence microscopy images for TDP-43 LCD (5 µM) in the presence of different concentrations of bis-ANS. **c** Representative turbidity traces for multiple protein concentrations as a function of bis-ANS to TDP-43 LCD molar ratio ($n \geq 3$ technical replicates across three separate preparations for 5 µM protein, $n = 3$ technical replicates for 2.5 and 10 µM). **d** Turbidity contour plot (phase diagram) depicting the relationship between bis-ANS and TDP-43 LCD concentration. Data points used for contouring are shown in black. **e** Colocalization fluorescence microscopy images of liquid droplets formed by TDP-43 LCD in the presence of bis-ANS at a ratio of 2:1. Protein solution was doped with Alexa Fluor 594-labeled TDP-43 LCD (100 nM). **f** Representative fusion events between TDP-43 liquid droplets formed in the presence of bis-ANS. Experiments in (**e**, **f**) were performed using untagged (shown) and His-tagged protein with similar results. **g** FRAP of bis-ANS-containing droplets ($n \geq 14$ individual droplets across three independent experiments) depicting rapid recovery within 45 s. Experiments were performed immediately after droplet formation and at 3 h. **h** Droplets formed by bis-ANS and TDP-43 LCD are disrupted at high temperatures ($n = 3$ technical replicates). All experiments were performed using a 20 mM potassium phosphate buffer, pH 6.0. Scale bars: 10 µm. All data are presented as mean values ± standard deviation.

partition together into the droplet phase (Fig. 1e). The droplets had liquid-like character, as indicated by fusion events that could be observed by time-lapse microscopy (Fig. 1f) and by fluorescence recovery after photobleaching (Fig. 1g). In stark contrast to homotypic droplets that generally mature within 1 h as demonstrated by others[6] and our lab[26] previously, droplets formed in the presence of bis-ANS maintained dynamicity for at least 3 h. Additionally, droplets displayed upper critical solution temperature (UCST) behavior (i.e., gradually disappeared with an increase in temperature, Fig. 1h)[38]. All of the experiments in this section were performed using TDP-43 LCD without any tag; however, we found that the presence of protein with a His-tag showed a similar behavior, even though the

condensation−decondensation curve was shifted to higher bis-ANS concentrations (Supplementary Fig. 2).

**Charge-based interactions drive heterotypic TDP-43 LCD LLPS.** As demonstrated by the phase diagram in Supplementary Fig. 1 as well as in previous literature[6,26,35,36], classical salt-induced LLPS involving homotypic TDP-43 LCD−LCD interactions does not exhibit such a condensation−decondensation phenomenon, starkly contrasting the phase diagram shown in Fig. 1d. In fact, the biphasic phase transition behavior found here for the bis-ANS/TDP-43 LCD system is rather reminiscent of that observed for protein/RNA systems. Even though bis-ANS is a

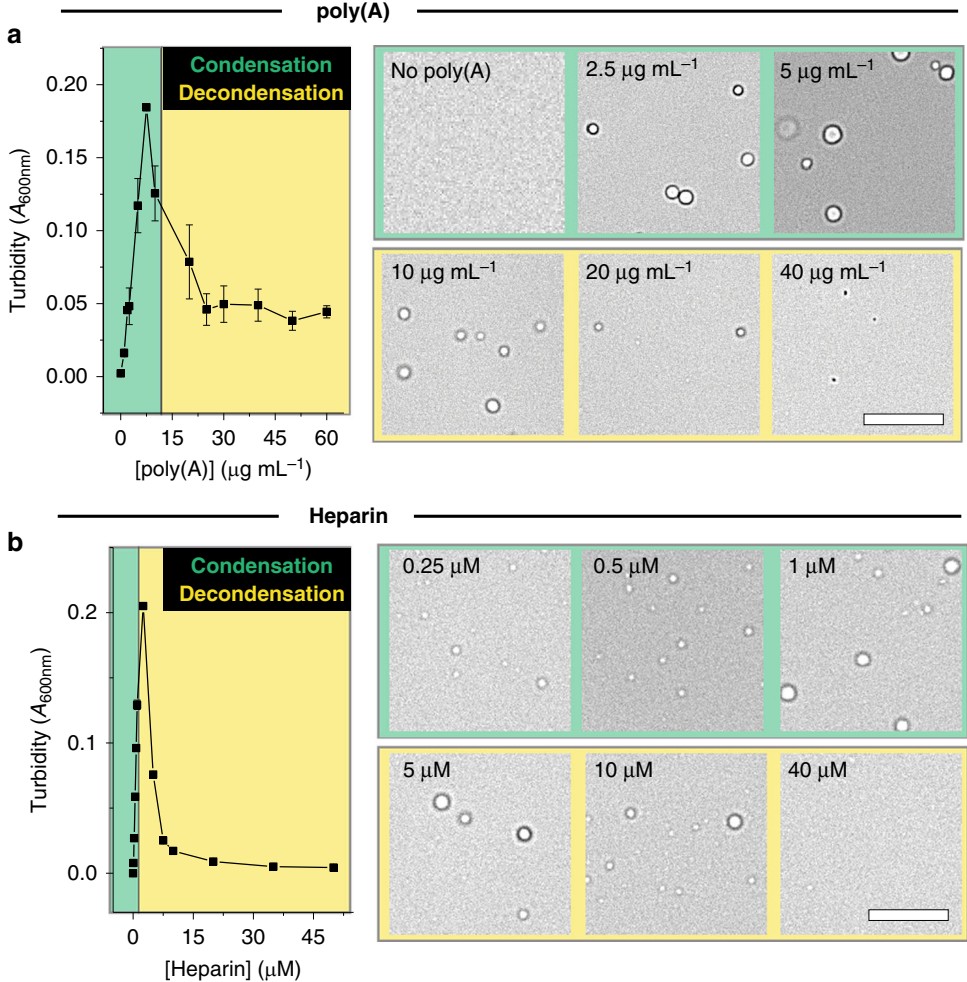

**Fig. 2 Poly(A) and heparin induce LLPS and a reentrant phase transition with TDP-43 LCD. a** Turbidity of TDP-43 LCD as a function of poly(A) concentration (left) and representative bright field microscopy images of droplets in the presence of different concentrations of poly(A) (right), indicating that low concentrations of poly(A) promote condensation while high concentrations result in decondensation. **b** Turbidity of TDP-43 LCD as a function of heparin concentration (left) and representative bright field microscopy images of droplets in the presence of different concentrations of heparin (right). Experiments were performed using 5 μM TDP-43 LCD in a 20 mM potassium phosphate buffer, pH 6.0. Scale bars: 10 μm. ($n \geq 3$ technical replicates). All data are presented as mean values ± standard deviation.

small molecule, it contains negative charges in addition to aromatic groups (Fig. 1a), akin to the basic unit of nucleotide polymers, such as RNA. These types of groups can engage in droplet-stabilizing ionic charge−charge or cation−π interactions with charged groups on protein[19]. Indeed, the TDP-43 LCD has been shown to undergo LLPS with RNA[6], though the mechanism underlying this phenomenon remains largely unknown. However, positively charged intrinsically disordered protein regions are commonly capable of interacting with RNAs[39] and the interaction between RNA and the TDP-43 C-terminal fragment (which contains the LCD) has been shown to reduce cytotoxicity[40]. Thus, we considered whether the observed biphasic regulation of LLPS by bis-ANS (Fig. 1c, d) was related to charge-based interactions between the two species, similar to protein/RNA systems.

To address this issue, we first explored as a point of reference the phase behavior of TDP-43 LCD in the presence of a model nucleotide polymer, poly(A), which could potentially engage in electrostatic interactions with the few lysine and arginine amino acids present in the TDP-43 LCD to facilitate LLPS (Supplementary Fig. 3)[19]. Again, these experiments were carried out at 5 μM TDP-43 LCD, which is well below the $c_{sat}$ for homotypic (salt-induced) phase separation (Supplementary Fig. 1). Titration of TDP-43 LCD

with poly(A) resulted in a biphasic condensation−decondensation curve, with a reentrant phase transition confirmed by bright field microscopy (Fig. 2a). Consistent with observations for other peptide/RNA systems[14,19], we found that the decondensation regime correlated with a large change in electrophoretic mobility ($\mu_e$) toward negative values, suggesting that droplets are destabilized by a strong, negative electrostatic repulsion (that is to say, neutrally charged condensates are destabilized by excess negative charge brought about by poly(A)[14]) (Supplementary Fig. 4a). Most recent models[19] describing such a transition in $\mu_e$ indicate that arginine amino acids can engage in cation−π interactions with RNA in the decondensation regime. Because TDP-43 LCD also contains arginine amino acids (Supplementary Fig. 3), a significant contribution of short-ranged attractive forces may also be present that therefore partially balance against this electrostatic repulsion that drives decondensation.

The potential involvement of cation−π interactions in peptide/RNA systems indicates that such systems are not typical complex coacervates, which, by definition, are based on ionic charge−charge interactions between oppositely charged polymers[41]. Therefore, we additionally explored the phase behavior of TDP-43 LCD in the presence of heparin, a negatively charged polymer that would

primarily engage in ionic charge−charge interactions. Again, titration of TDP-43 LCD with heparin resulted in a biphasic condensation−decondensation curve (Fig. 2b). These data revealed that the TDP-43 LCD can engage in electrostatic interactions with anionic polymers, possibly through a combination of ionic charge−charge and cation−π interactions, depending on the cofactor used.

Importantly, both types of charge-based interactions can be disrupted through the addition of salt[17,20,42]. Indeed, the addition of high concentrations of NaCl (>150−200 mM) completely disrupts the ability for the TDP-43 LCD and poly(A) mixture to undergo LLPS (Fig. 3a, green panel) and such a disruption was further verified by bright field microscopy (Supplementary Fig 5a, green panel). A similar disruption by NaCl occurred in the presence of heparin (Fig. 3b, green panel, and Supplementary Fig. 5b, green panel). Both of these experiments were initially performed using poly(A) or heparin concentrations coinciding with the condensation regime (as originally defined in the absence of NaCl in Fig. 2). When NaCl was added under conditions coinciding with the decondensation regime, a different behavior was observed: low NaCl concentrations (up to ~25 mM) resulted in an increased turbidity, while higher NaCl concentrations subsequently caused turbidity to diminish to zero (Fig. 3a, b, yellow panels). Thus, it appears that the addition of small concentrations of salt to poly(A)/TDP-43 LCD or heparin/TDP-43 LCD systems under conditions that would typically be associated with the decondensation regime can re-stabilize droplet formation, and this was further confirmed by bright field microscopy (Supplementary Fig. 5a, b, yellow panels). Altogether, these findings suggest that low salt concentrations neutralize the electrostatic repulsion brought about by poly(A)- or heparin-mediated protein charge overscreening, possibly without disrupting attractive forces. However, higher concentrations of salt disrupt these attractive forces and fully destabilize droplets (Supplementary Fig. 5a, b, yellow panels), akin to what is observed when adding salt under conditions coinciding with the condensation regime (Fig. 3a, b, green panels).

**The role of charge in bis-ANS-induced phase transitions**. Remarkably, in sharp contrast to the strong salt dependence of poly(A)/TDP-43 LCD and heparin/TDP-43 LCD droplets (Fig. 3a, b, green panels), the addition of NaCl (even up to 500 mM) had very little disruptive effect on bis-ANS-induced LLPS (Fig. 3c, green panel; Supplementary Fig. 5c, green panel). This suggests that the interactions between bis-ANS and TDP-43 LCD that lead to induction of LLPS (the condensation regime) are largely nonelectrostatic in nature.

Under conditions corresponding to the decondensation regime, low salt concentrations (25−75 mM NaCl) caused an increase in turbidity, akin to the observations for the poly(A)/TDP-43 LCD and heparin/TDP-43 LCD systems (Fig. 3a−c, yellow panels). This indicates that, in all of these systems, electrostatic repulsion (due to the negative charges on poly(A), heparin or bis-ANS) likely drives the decondensation. For the bis-ANS/TDP-43 LCD system, this notion is further supported by $\mu_e$ experiments, which revealed a shift from a slightly positive to a large negative $\mu_e$ as bis-ANS concentration was increased (Supplementary Fig. 4b). However, in sharp contrast to the polyanion/TDP-43 LCD systems, bis-ANS/TDP-43 LCD droplets were not disrupted by further increases in NaCl concentration (Fig. 3c, yellow panel, and Supplementary Fig. 5c, yellow panel). Thus, in this case, higher salt concentrations strongly inhibit the reentrant phase transition. Altogether, these data indicate that the reentry mechanisms in all three systems are generally similar, based on repulsive electrostatic interactions between protein-associated cofactor molecules. This

notion is consistent with the observation that addition of increasing concentrations of bis-ANS to the poly(A)/TDP-43 LCD system at the poly(A) to protein ratio corresponding to maximum turbidity results in a reentrant phase transition, most likely due to the presence of additional negative charges from the bis-ANS molecules (Supplementary Fig. 4c).

However, the unique persistence of bis-ANS/TDP-43 LCD droplets at higher salt concentrations (both in the condensation and decondensation regimes) suggests that the mechanism of protein interaction with bis-ANS might be fundamentally different from the charge-based mechanism responsible for interactions with poly(A) and heparin[19,41]. To explore this issue, we probed the effect of ionic strength on the interaction between bis-ANS and TDP-43 LCD. To this end, we used a fluorescence resonance energy transfer (FRET)-based method[43] by exciting the tryptophan residues at 280 nm and recording emission spectra in the presence of varying concentrations of bis-ANS (Fig. 3d). In contrast to other fluorescence-based methods for studying binding, such as anisotropy, FRET would be less prone to changes in viscosity related to liquid droplet formation and would therefore be a more practical approach over a range of bis-ANS concentrations. Importantly, the binding curves obtained by plotting tryptophan fluorescence at 356 nm as a function of bis-ANS concentration were essentially unaffected by the presence of salt (Fig. 3e), strongly indicating that the interactions between bis-ANS and TDP-43 are not charge-based in nature.

When interpreting the above experiments, it is important to recognize that, at the protein concentrations tested here (5 μM), salt-induced homotypic interactions in the absence of cofactors do not become significant enough to induce LLPS below ~1 M NaCl (Supplementary Fig. 1a). While we cannot fully exclude the possibility that bis-ANS may interact electrostatically to alter the propensity for such homotypic interactions, it remains unclear why this would only be the case for bis-ANS-induced LLPS, but not heparin- or poly(A)-induced LLPS (the latter of which are strongly based on electrostatic interactions).

**The role of bivalence and hydrophobicity in bis-ANS-induced LLPS**. To gain more detailed insight into the mechanism by which bis-ANS induces LLPS of the TDP-43 LCD, we used turbidity measurements to compare the LLPS-promoting capacity of a library of chemical compounds. Many compounds tested represent a specific building block of the overall bis-ANS chemical structure (Fig. 4a). For instance, bis-ANS is a dimeric analog of 8-anilino-1-sulfonic acid (ANS, **2**). Therefore, we first tested whether the bivalent nature of bis-ANS was important to its interaction with the TDP-43 LCD. While bis-ANS could facilitate LLPS at concentrations below 5 μM, two orders of magnitude higher concentrations of monovalent ANS were required for LLPS (Fig. 4b). Droplets formed in the presence of ANS appeared spherical (Fig. 4) and partial fusion events could be observed (Supplementary Fig. 6a), indicating liquid-like character. This suggested that the bivalent nature of bis-ANS is an important feature, even though likely not sufficient as ANS could still induce LLPS but at much higher concentrations. We next considered testing bivalent naphthalenes with no charged groups or positively charged groups. However, the uncharged 1,1′-binaphthyl (**3**; Supplementary Fig. 6b) had extremely low solubility in aqueous buffer and naphthyl-based quaternary amines (which would be the most likely candidates for testing positively charged naphthalene derivatives) have been shown to form micelles[44]. Thus, these features precluded meaningful measurements with both types of compounds.

To combat this inability for testing other bivalent naphthalenes, we examined the ability for non-naphthalene, bivalent, and

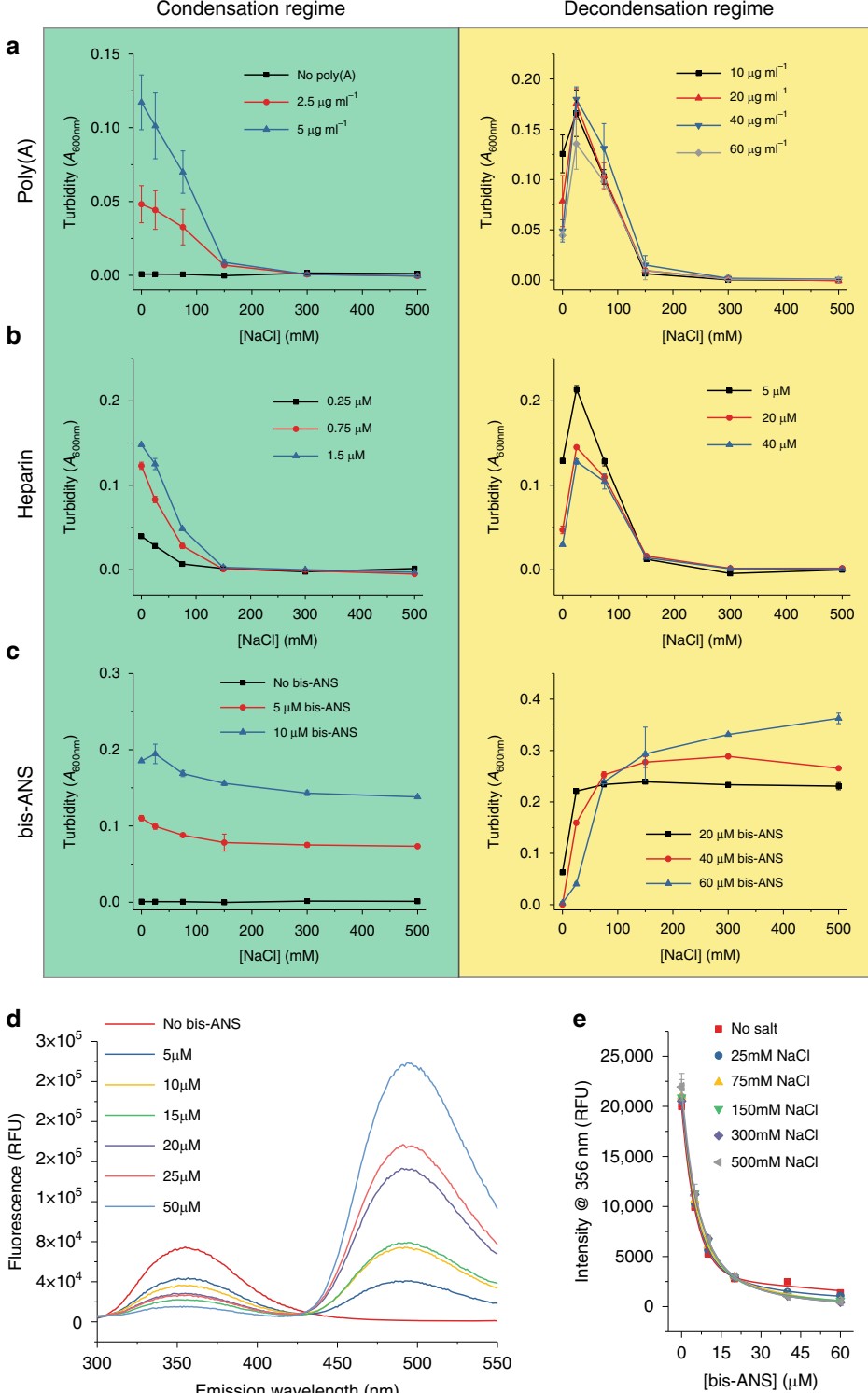

**Fig. 3 Role of charge-based interactions in various TDP-43 LLPS systems. a–c** Turbidity of TDP-43 LCD (5 µM) as a function of salt concentration in the presence of varying cofactor concentrations, including **a** poly(A), **b** heparin, and **c** bis-ANS. Individual columns depict data for cofactor concentrations associated with the condensation regime (green) or decondensation regime (yellow) for biphasic curves as originally defined in the absence of NaCl ($n \geq 3$ technical replicates). **d** Fluorescence spectra (280 nm excitation wavelength) for TDP-43 LCD (10 µM) in the presence of varying concentrations of bis-ANS. **e** Fluorescence intensity at 356 nm for TDP-43 LCD as a function of bis-ANS concentration in the presence of varying concentrations of NaCl. All experiments depicted in this figure were performed in 20 mM potassium phosphate buffer, pH 6.0, using untagged protein at a concentration of 5 µM ($n = 3$ technical replicates). All data are presented as mean values ± standard deviation.

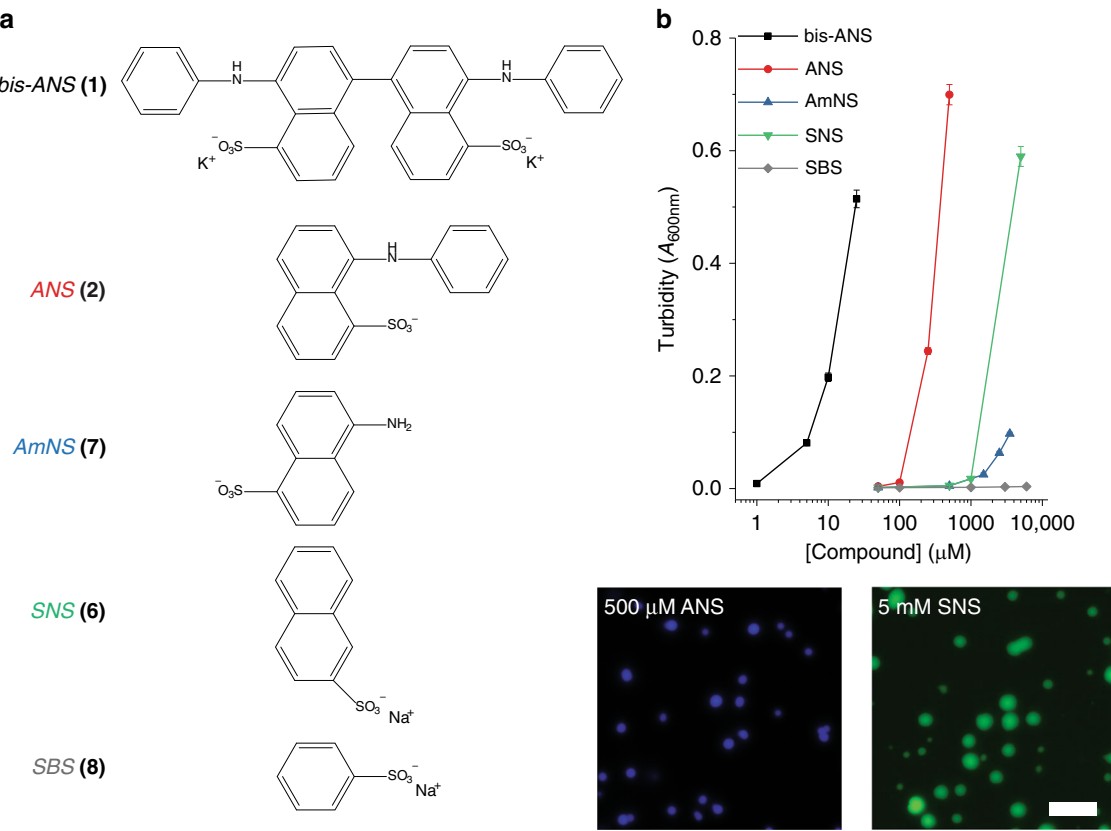

**Fig. 4 Comparison of the ability for naphthalene sulfonate derivatives to induce TDP-43 LCD LLPS. a** Chemical structures of naphthalene derivatives: bis-ANS (**1**), 8-anilinonaphthalene-1-sulfonic acid (ANS, **2**), 5-amino-1-naphthalenesulfonic acid (AmNS, **7**), sodium 2-naphthalene sulfonate (SNS, **6**), and sodium benzene sulfonate (SBS, **8**). **b** Turbidity as a function of compound concentration for chemicals depicted in panel (**a**) (top panel) and representative fluorescence microscopy images of droplets formed in the presence of ANS or SNS (bottom panels). Droplets containing SNS were imaged using protein doped with Alexa Fluor 488-labeled TDP-43 LCD. Experiments were performed in a 20 mM potassium phosphate buffer, pH 6.0, using 10 µM His-tagged TDP-43 LCD. Scale bar: 10 µm. All data are presented as mean values ± standard deviation ($n = 3$ technical replicates).

negatively charged molecules with and without polycyclic aromaticity (cyclic di-AMP (**4**) and heparin disaccharide I-H (**5**), respectively) to induce LLPS. While nucleotide polymers (i.e., poly(A)) can induce LLPS of TDP-43 LCD (Fig. 2a), the bivalent adenine-containing cyclic di-AMP could not (Supplementary Fig. 6c). Likewise, while a large heparin polymer can readily induce LLPS of the TDP-43 LCD, its dimeric analog cannot (Supplementary Fig. 6d). Furthermore, the heterodimeric nicotinamide adenine dinucleotide (NAD) did not induce LLPS over the concentrations tested (Supplementary Fig. 6e). These findings again revealed that bivalence in and of itself is not solely sufficient for promoting LLPS of the TDP-43 LCD. It should be noted here that recent reports have indicated that ATP can have a modulatory effect on LLPS of some proteins, showing enhancement of previously present LLPS at low concentrations and disruption of droplets at higher concentrations[45–48]. However, unlike naphthalene sulfonate-based compounds, no de novo induction of LLPS by ATP has been observed and any modulatory effects require millimolar ATP concentrations. Consistent with these reports, we observed no ability for ATP (or cyclic AMP) to induce LLPS de novo for the TDP-43 LCD, even through 10 mM concentrations for ATP (Supplementary Fig. 6f, g). These findings indicate that the ability to induce LLPS de novo is not shared across many small molecules, but may result from the specific chemical nature of naphthalene derivatives.

Importantly, the log$P$ values (partition coefficients between polar and nonpolar solvents) for adenosine and ANS are −2.367 and 0.952, respectively (www.chemspider.com), indicating that naphthalene sulfonates have much stronger hydrophobic character (Supplementary Table 1). Indeed, droplets formed by TDP-43 LCD in the presence of bis-ANS were readily disrupted by the addition of 1,6-hexanediol, which can disrupt hydrophobic interactions[49] (Supplementary Fig. 7). We therefore further explored monovalent naphthalene sulfonate derivatives in our library to determine whether hydrophobicity was important for facilitating such a strong LLPS-inducing effect of bis-ANS. The compounds we tested all contained the negatively charged sulfonate group (removal of which would render them insoluble in aqueous buffer) and therefore only differ in hydrophobic cores. SNS (sodium 2-naphthalene sulfonate, **6**)—which lacks the hydrophobic aniline group of ANS—required two-fold higher concentrations (~1 mM) to induce LLPS as compared to ANS (Fig. 4a, b). The presence of a primary amine did not appear to play a significant role, as both SNS and AmNS (5-amino-1-naphthalenesulfonic acid, **7**) began to promote LLPS at around the same concentrations (Fig. 4b). Subsequent time-lapse microscopy revealed that fusion events occurred between droplets formed by the addition of SNS (Supplementary Fig. 6h), confirming their liquid-like nature. Finally, the loss of polycyclic character appeared to disable the capacity for LLPS induction, as SBS (sodium benzene sulfonate, **8**) was unable to promote LLPS at concentrations through 5 mM. Cumulatively, these data strongly suggest that the induction of TDP-43 LCD LLPS by bis-ANS is mediated by the combination of two intrinsic features of bis-ANS: its bivalence and the presence of highly hydrophobic naphthalene groups.

**Identification of another LLPS-modulating compound.** The insights gained from studies with bis-ANS might reflect general chemical principles or criteria that could be utilized for identification of other LLPS-modulating compounds. To test this, we searched for another molecule with the following characteristics: bivalence, highly hydrophobic moieties (i.e., naphthalene), and negatively charged groups. The last of these criteria is essential for the electrostatic repulsion that drives decondensation. This led us to Congo red (CR, **9**), a dye commonly employed to identify amyloid aggregates in patient brains[33,50] (Fig. 5a). Akin to bis-ANS, addition of low concentrations of CR strongly promoted LLPS of the TDP-43 LCD as evidenced by turbidity measurements and bright field microscopy (Fig. 5b, c). Condensates formed by TDP-43 LCD and CR displayed liquid-like character as well, in which fusion events could be observed by microscopy (Fig. 5d). Furthermore, higher concentrations of CR also resulted in a reentrant phase transition (Fig. 5b). These results validate that the insights gained from our studies with bis-ANS and related naphthalene derivatives can be used to identify other small molecule modulators of LLPS.

**Bis-ANS modulates the phase behavior of many proteins.** Our findings thus far were limited to studies using the TDP-43 LCD. We therefore asked whether our observations with bis-ANS could represent a general approach to modulating LLPS of other LCDs and full-length proteins. We first began with the FUS LCD, which undergoes homotypic LLPS mediated by repetitive SYGQ-rich

motifs[51]. Indeed, a similar condensation and reentrant phase transition was observed (Fig. 6a). We next investigated the DEAD-box RNA helicase, Ded1p, which is the *Saccharomyces cerevisiae* ortholog of human DDX3X[52]. Ded1p is a key regulator of MLOs and undergoes LLPS in vitro[53]. Under conditions at which no LLPS was observed for protein alone (1 μM Ded1p), bis-ANS could induce de novo condensation for the full-length protein (Fig. 6b). Furthermore, a reentrant phase transition could be observed at higher concentrations of bis-ANS, akin to studies with TDP-43 and FUS LCDs. We continued our investigation by employing the microtubule-binding protein tau, which forms aggregates in neurodegenerative diseases[54,55] and has recently been shown to undergo LLPS[13,56–58]. While in vitro LLPS of tau alone requires the presence of crowding agents such as poly-ethylene glycol or dextran[56,58], bis-ANS could induce robust condensation of tau in the absence of any crowder (Fig. 6c). Again, a reentrant phase transition occurred at higher concentrations of bis-ANS. These findings demonstrate that the modulatory capacity of bis-ANS is not only limited to the TDP-43 LCD, but also extends to other proteins.

**Bis-ANS inhibits homotypic LLPS in a crowded environment.** While higher concentrations of bis-ANS can facilitate a reentrant phase transition for droplets formed in the presence of lower concentrations of bis-ANS, it was not yet clear whether this modulatory effect could extend to and regulate LLPS mediated by homotypic interactions occurring in the crowded cellular

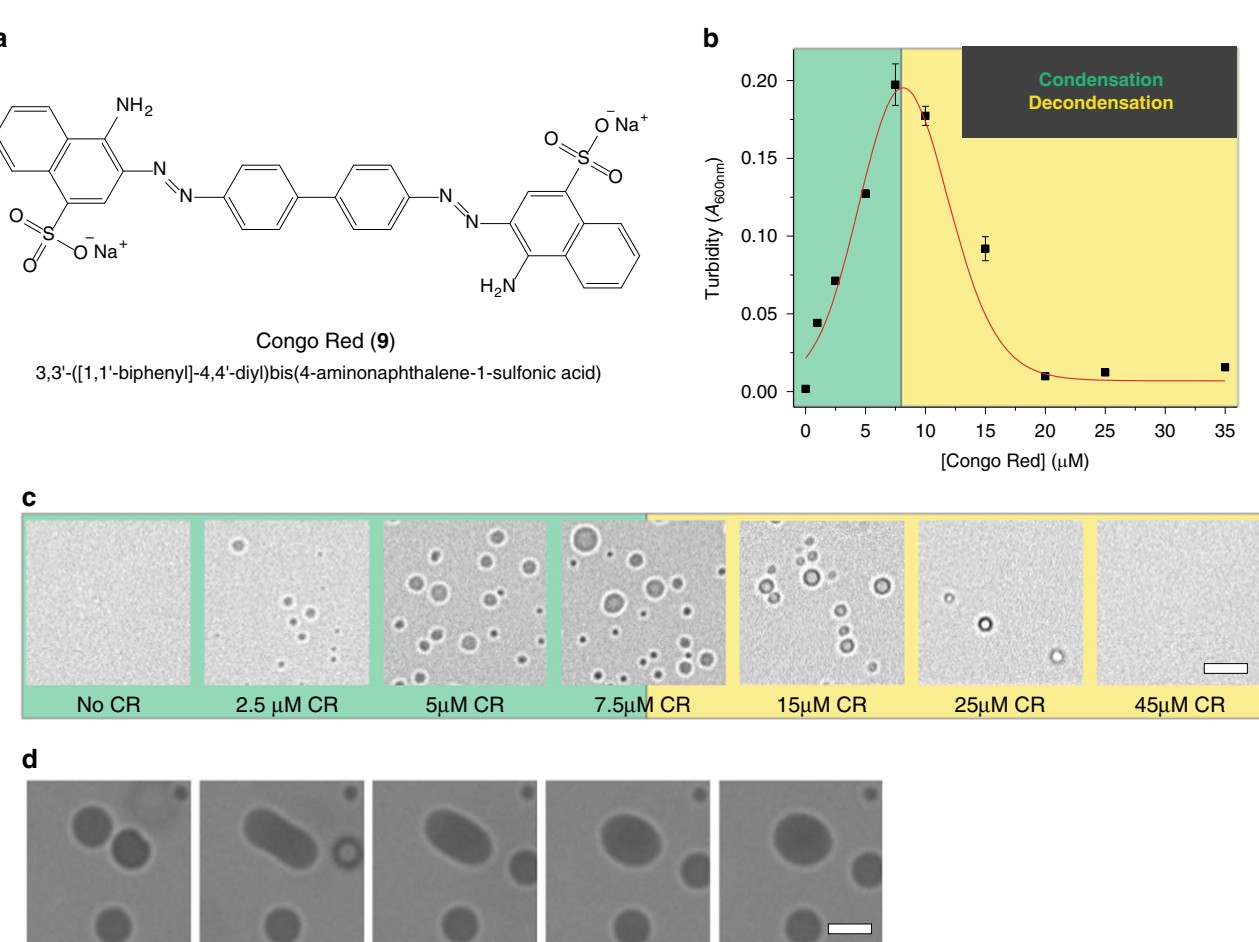

**Fig. 5 Congo red modulates the phase behavior of the TDP-43 LCD. a** Chemical structure of Congo red (CR, **9**). **b** Turbidity of TDP-43 LCD (untagged; 5 μM) as a function of Congo red concentration. **c** Representative bright field microscopy images of CR/TDP-43 LCD system. **d** Representative images of CR/TDP-43 LCD droplet fusion event (molar ratio of CR to TDP-43 of 1.3:1). Scale bars: 5 μm. Experiments were performed in a 20 mM potassium phosphate buffer, pH 6.0. All data are presented as mean values ± standard deviation (*n* = 3 technical replicates).

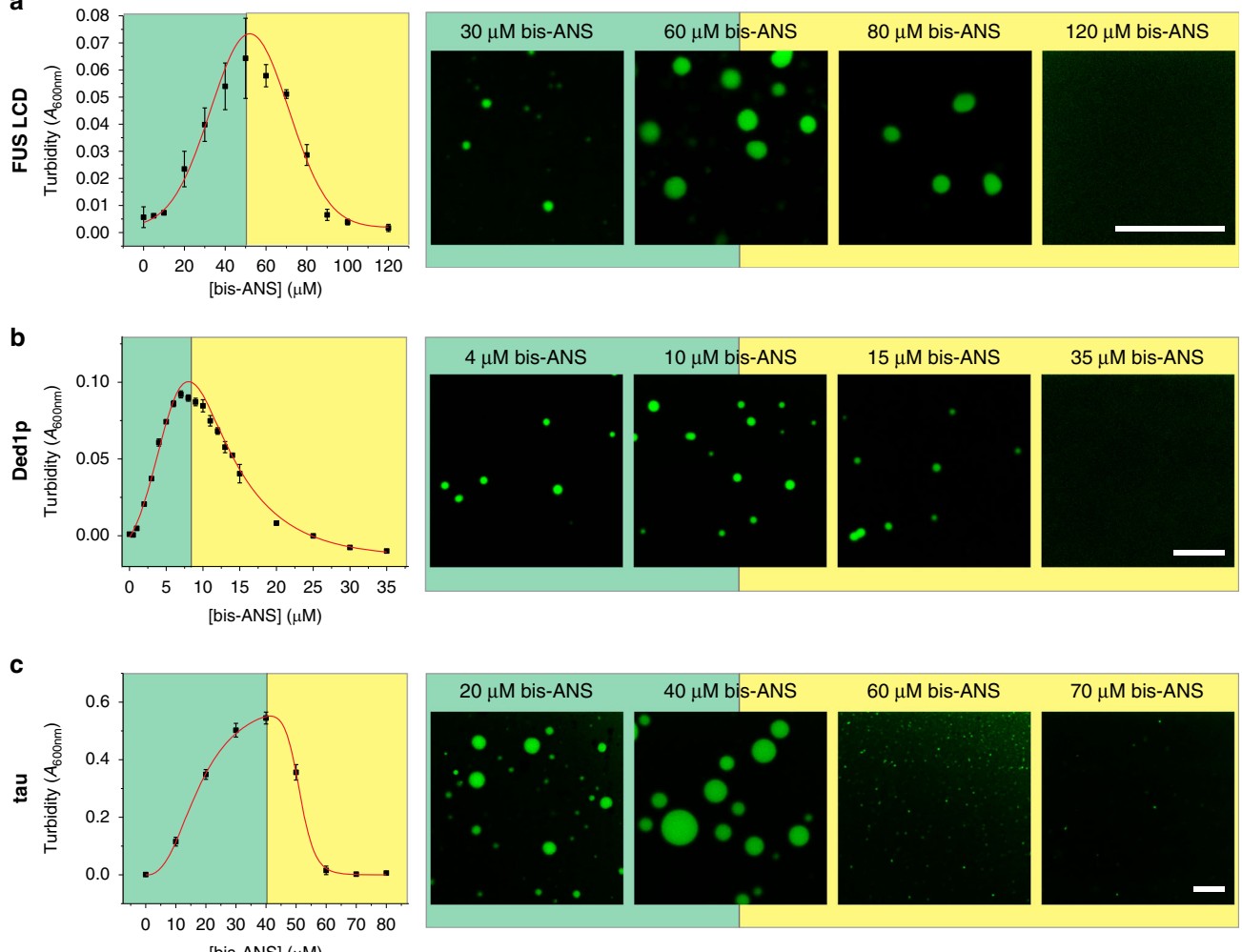

**Fig. 6 Bis-ANS modulates phase behavior of FUS LCD, Ded1p, and tau. a, b** Turbidity curves and representative fluorescence microscopy images of **a** FUS LCD (20 μM), **b** full-length Ded1p (1 μM) and **c** full-length tau (10 μM) in the presence of varying bis-ANS concentrations. Scale bars: 10 μm. All data are presented as mean values ± standard deviation ($n = 3$ technical replicates).

environment. LLPS mediated by these interactions are strongly associated with the formation of pathological aggregates, such as amyloid fibrils[24,56,59]. Thus, the ability to prevent these potentially aberrant phase transitions would parallel the capacity for some molecules, like ATP, to act as hydrotropes[48]. Homotypic LLPS that occurs in the crowded cellular environment can be replicated in the test tube through the addition of crowding agents, such as dextran or polyethylene glycol[26,56,58]. Therefore, we tested whether bis-ANS can also prevent the formation of TDP-43 LCD droplets under conditions that strongly favor homotypic LLPS (i.e., in the presence of crowding agent) without any other cofactors. Again, high concentrations of bis-ANS completely abrogated droplet formation by TDP-43 LCD (Supplementary Fig. 8a). Furthermore, bis-ANS was also able to dissolve preexisting TDP-43 LCD droplets, even 30 min after formation (Supplementary Fig. 9). Importantly, this inhibitory effect of bis-ANS on homotypic LLPS in a crowded environment was also observed for a number of full-length proteins, including Ded1p, FUS and tau (Supplementary Fig. 8b−d).

Testing the effect of bis-ANS on LLPS of full-length TDP-43 was a little more complicated, as the latter protein is highly aggregation prone. Therefore, we employed the approach of Wang et al.[60], in which cleavage of the maltose binding protein solubility tag fused to the full-length TDP-43 leads to transient LLPS, followed by relatively fast protein aggregation. The addition of bis-ANS at higher concentrations completely abrogated the ability for the full-length TDP-43 to form droplets (and subsequently also prevented aggregation of the protein) (Supplementary Fig. 8e). Altogether, these data demonstrate that not only can bis-ANS (at very low concentrations) facilitate protein droplet formation de novo, but also that it can (at somewhat higher concentrations) prevent LLPS that is associated with protein aggregation.

**Bis-ANS modulates stress granule dynamics in cells.** We next wanted to determine whether bis-ANS could potentially modulate LLPS in a cellular setting. To address this question, we first investigated whether bis-ANS was cytotoxic in an HCT-116 colon cancer cell line. Dose-dependent bis-ANS toxicity was only observed above 400 μM concentrations after treatment for 6 and 24 h (Fig. 7a), indicating that the molecule was nontoxic under conditions where it might be used in cell treatments. Cumulatively, our in vitro investigations thus far had revealed that bis-ANS can be used both as a modulator of LLPS and as a marker of droplets due to the molecule's intrinsic fluorescence. However, live imaging of cells treated with bis-ANS revealed that the molecule robustly stains many intracellular structures, likely due to its affinity for hydrophobic moieties (Fig. 7b). We therefore

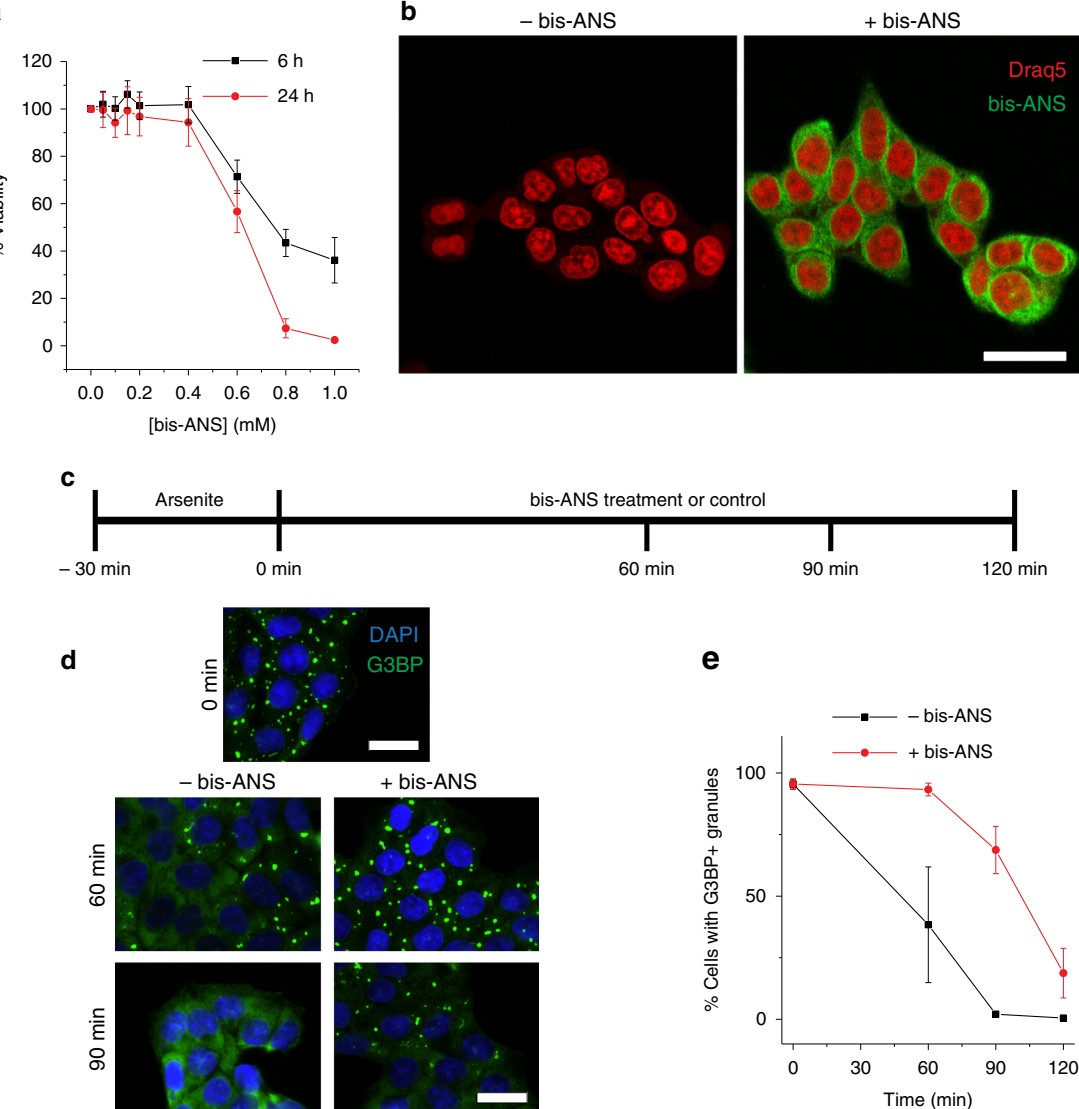

**Fig. 7 Bis-ANS modulates stress granule dynamics in cells. a** HCT-116 cell viability upon treatment with increasing concentrations of bis-ANS for 6 or 24 h as determined by 3-(4,5-dimethylthiazol-2-yl)-2,5-diphenyltetrazolium bromide (MTT) assay. Error bars represent SD ($n = 5$ technical replicates across two biological replicates). **b** Live-cell imaging of HCT-116 cells with or without 1 h treatment with 250 μM bis-ANS. Nuclei were stained with 1:1000 Draq5. **c** Schematic diagram of sodium arsenite washout experiments described. HCT-116 cells were treated for 30 min with 500 μM sodium arsenite in culture medium. Arsenite was removed and medium containing 250 μM bis-ANS or medium alone (control) was added. Cells at different time points after sodium arsenite removal were fixed and stained with DAPI and with an antibody against G3BP. **d** Representative microscopy images obtained 60 and 90 min after arsenite removal. **e** Quantitative assessment of percentage of cells exhibiting punctate G3BP staining in the presence and absence of bis-ANS at different times after arsenite removal. Note that data points at 0 min represent analysis of the same images for control and treatment groups. Scale bars represent 20 μm. All data are presented as mean values ± standard deviation ($n = 3$, each independent experiment analyzing 20 fields of view).

surmised that assessing granule dynamics in the cellular setting by monitoring bis-ANS fluorescence would not be reliable, because the staining and modulation of granules by bis-ANS could not be readily distinguished from each other, in addition to the surrounding background fluorescence.

We therefore employed an immunofluorescence approach to test whether bis-ANS could modulate stress granules in a cell culture model. To this end, HCT-116 cells were treated with sodium arsenite for 30 min in order to induce G3BP+ stress granules. Arsenite was then removed (or washed out) and medium was added with or without bis-ANS. During this washout, cells were periodically fixed and stained with a G3BP-reactive antibody (a commonly used marker for stress granules[61]) in order to monitor the dissolution of stress granules over time. Typically, G3BP+ stress granules dissolve within 90 min in the absence of

any other molecules (Fig. 7d, e). However, when bis-ANS was present after the removal of arsenite, G3BP+ stress granules were still observed in cells for up to 2 h (Fig. 7d, e). In the absence of sodium arsenite pretreatment, bis-ANS under the conditions used was unable to induce stress granules de novo (Supplementary Fig. 10). This observation suggests that bis-ANS is not an inherent stressor of cells (in contrast to arsenite), but rather a modulator of stress granules. Taken together, these data demonstrate that bis-ANS could be a prototype for small molecules that have the capacity to act as modulators of LLPS in cells.

## Discussion
In light of the finding that dysregulated LLPS can facilitate the formation of deleterious protein aggregates[26,59,62], there has been great interest in identifying compounds that might modulate

LLPS. Efforts have been primarily aimed at finding small molecules that can disrupt aberrant phase transitions to protein aggregates by preventing LLPS[28,29]. However, a growing body of evidence indicates that LLPS is also crucial to the normal functioning of the cell[4,31,32]. Therefore, finding molecules that can minimize aberrant phase transitions while maintaining physiological biomolecular condensation is of equal—if not even greater—importance. In this work, we have identified a class of small molecules, including bis-ANS and Congo red, as potent modulators of protein LLPS and elucidated the mechanism of this modulatory action by using TDP-43 LCD as a model protein. Not only can low concentrations of these molecules initiate LLPS de novo, but also higher concentrations result in decondensation of liquid droplets. Such biphasic behavior is reminiscent of the regulatory capacity of protein/RNA condensates in the cell[21].

In an effort to better understand the interaction between TDP-43 LCD and bis-ANS, we revealed insights about the protein's ability to form liquid droplets in the presence of negatively charged polymers, poly(A) and heparin. Although the TDP-43 LCD has been shown previously to interact with RNA to undergo LLPS[6], such a finding was somewhat peculiar, considering the protein is largely comprised of hydrophobic or uncharged polar regions and has a modest net positive charge (+3) (Supplementary Fig. 3). TDP-43 LCD contains only five arginine and one lysine residues, which are important for charge-based interactions in other protein systems[19]. Furthermore, there is only one RGG segment, a motif associated with disordered protein−RNA interactions in LLPS systems[63]. However, other repetitive, glycine-containing sequences, some of which can be found in TDP-43 LCD, may potentially modulate protein−RNA and protein−poly(A) interactions[63]. These include FG (of which the TDP-43 LCD has six, one of which is FGG) and a single PGG motif (Supplementary Fig. 3). Additionally, the LCD contains segments rich in polar glutamine residues, which have been shown to interact with RNA and undergo LLPS[64].

While there is still much to learn about the molecular basis of TDP-43 LCD interactions with poly(A) and heparin, the ability for salt to disrupt droplet formation clearly demonstrates that condensate stability is mediated by charge-based interactions. Because both poly(A) and heparin can only contribute negative charges, the positive charge must come from the TDP-43 LCD. In stark contrast to these systems, droplet formation by bis-ANS and TDP-43 LCD remains largely unimpeded by the presence of NaCl, which indicates that the interactions driving LLPS in this system are not largely based on charge. This is further supported by FRET experiments which reveal that binding between bis-ANS and TDP-43 LCD is unaffected by the addition of salt. Altogether, these data provide evidence that bis-ANS/TDP-43 LCD droplets are fundamentally distinct from condensates formed in the presence of poly(A) and heparin.

In order to better understand the mechanism by which bis-ANS and Congo red modulate LLPS of the TDP-43 LCD, we screened a small library of compounds similar to bis-ANS, which revealed that only those that contain naphthalene group(s) (i.e., bis-ANS, ANS, AmNS, SNS) are capable of inducing LLPS. As suggested by logP values (Supplementary Table 1), such moieties most likely interact hydrophobically with protein and possibly via π−π stacking. In this context, it is important to note that bis-ANS-induced droplets do display UCST behavior. While LLPS systems mediated by hydrophobic interactions (i.e. elastin-like peptides, which are highly hydrophobic and behave similarly to synthetic polymers[65]) are classically associated with lower critical solution temperature (LCST) behavior, there are certainly exceptions. For example, tau protein/RNA condensates, which involve charge-based interactions, display LCST behavior[13]. Conversely, homotypic TDP-43 LLPS, which is mediated by hydrophobic

interactions, displays UCST behavior[6,26,36]. Such ambiguity likely reflects the complex interactions at play in protein-mediated LLPS. Furthermore, the negative charge brought by sulfonate groups may play a role in charge neutralization. This possibility is supported by the finding that the presence of a positively charged His-tag on the TDP-43 LCD causes a shift in peak turbidity to higher ratios of bis-ANS to protein (Supplementary Fig. 2).

Remarkably, bis-ANS and Congo red stand out from all the other naphthalene sulfonate derivatives tested because they induce LLPS at much lower concentrations. Such a strikingly strong modulatory effect on LLPS appears to stem from the molecules' bivalent nature, which could allow them to act as small intermolecular scaffolds or transient, weak cross-linkers between protein molecules. This expands upon the concept that multivalent interactions are an important feature in phase-separating systems[1,10,66], but further implies that a dimeric small molecule can bring about sufficient valence for condensate formation by a disordered protein region. Importantly, this effect is not observed for less hydrophobic, bivalent compounds, including cyclic-di-AMP, heparin disaccharide I-H, and NAD, which indicates that the strength of interaction between the small molecule's functional group (i.e., naphthalene) and the protein are critical determinants of such activity. While ionic charge−charge and short-range cation−π interactions are important for stabilizing peptide/RNA condensates[19], this appears to only be sufficient for LLPS when in the polymeric form, as the dimeric analog, cyclic-di-AMP, does not induce LLPS. The same appears to be the case for heparin polymer and heparin disaccharide I-H. Altogether, the present results indicate that the ability to engage in strong hydrophobic interactions with proteins, bivalence, and the presence of negatively charged moieties (which are essential for the electrostatic repulsion that drives decondensation) are, in combination, key molecular features that allow small molecules to modulate LLPS of proteins. However, none of these characteristics can be considered exclusively sufficient to have this effect on their own.

Importantly, a similar modulatory effect by bis-ANS could be observed for several other proteins tested in this study, including tau, FUS (LCD and full-length), full-length TDP-43, and Ded1p. These findings suggest a lack of sequence specificity for the action of bis-ANS, as each of these proteins contains various combinations of polar, nonpolar, and charged residues as well as the presence or absence of repetitive motifs. Even more, robust homotypic LLPS for tau, TDP-43 (full-length and LCD), and FUS, which reportedly promotes disease-associated aggregation[24,56], could be prevented by the addition of bis-ANS. Accordingly, the specific chemical features and mechanisms revealed by our studies likely reflect general principles by which LLPS might be modulated by small molecules across multiple protein systems.

The main focus of this study was to elucidate the physicochemical properties of small molecules that have the capacity to modulate LLPS in vitro. Nevertheless, limited studies using a cell culture model revealed that bis-ANS is relatively nontoxic and, most importantly, can also modulate LLPS in mammalian cells by stabilizing stress granules. Decondensation of granules by bis-ANS was not observed in our present cellular model. This could be due to the high affinity of bis-ANS for many intracellular structures (as indicated by fluorescence microscopy images demonstrating promiscuous bis-ANS staining), resulting in insufficient concentrations of free bis-ANS to disrupt stress granules inside the cell. Further optimization of molecules of this class would certainly be needed in order to develop compounds of potential therapeutic value. This would also need to consider the complexity of MLOs in biological systems, which contain many different proteins and nucleic acids. The physicochemical

principles established in this study should guide efforts in this direction, providing a foundation for rational design of even more potent and selective small molecules that might effectively modulate LLPS in a cellular environment.

## Methods

**Protein expression and purification.** TDP-43 LCD with a His-tag and thrombin cleavage site was expressed and purified on an Ni-charged nitrilotriacetic acid (Ni-NTA) column as described previously[26]. For preparation of untagged TDP-43 LCD protein, thrombin cleavage conditions were optimized to minimize protein aggregation and maximize cleavage and recovery. To this end, Ni-NTA elution fractions were concentrated to ~7 mg mL$^{-1}$ and diluted 14–15-fold in a 20 mM potassium phosphate buffer, pH 6.0, containing thrombin (25 units mg$^{-1}$ of TDP-43 LCD). Cleavage was carried out at ambient temperature with mild stirring for 4–6 h. Fully cleaved protein (containing TDP-43 residues 266–414 with an N-terminal glycine) was confirmed by gel electrophoresis, solubilized in 6 M guanidine hydrochloride, and then concentrated to ~10 mg mL$^{-1}$. Untagged protein was separated from free His-tag and thrombin by HPLC using a C$_4$ column acetonitrile gradient in water containing 0.05% trifluoroacetic acid. Single-Cys variants for AlexaFluor™ 488 and AlexaFluor™ 594 labeling were generated as described previously[26]. Purified proteins (tagged or untagged) were flash frozen and lyophilized.

Recombinant full-length tau (residues 1–441) was expressed and purified according to the previously published procedure[58], with the exception that the dialysis step was omitted. Ded1p was purified as previously described[67]. The proteins were flash frozen and stored at −80 °C until use.

Vector for bacterial expression of full-length FUS protein with an N-terminal maltose binding protein (MBP) tag was a gift from Nicolas Fawzi (Addgene plasmid # 98651)[68]. The protein was expressed in Rosetta™ (DE3) pLysS competent cells (MilliporeSigma) upon induction with 1 mM IPTG. After 16 h growth at 15 °C, cells were harvested by centrifugation, suspended in lysis buffer (50 mM Tris, pH 7.4, 1 M NaCl, 10% glycerol, 20 mM imidazole, 1 mM phenylmethylsulfonyl fluoride (PMSF), 1 mM dithiothreitol (DTT)), bound to Ni-NTA resin, and washed with five column volumes. Protein was eluted using a linear gradient of 500 mM imidazole. Fractions containing FUS were pooled and mixed with amylose resin high flow (NEB). Bound protein was washed with ten column volumes of amylose binding buffer (50 mM Tris, pH 7.4, 1 M NaCl, 10% glycerol, 1 mM PMSF, 1 mM DTT). Protein was eluted with the same buffer but additionally containing 10 mM D(+)-maltose. Fractions were pooled and concentrated to 75 μM, flash frozen, and stored at −80 °C.

Vector containing FUS LCD (residues 1–165 in pRSET-b, codon optimized from BlueHeron) was sub-cloned from full-length FUS with an N-terminal His-tag. FUS was expressed in Rosetta™ (DE3) pLysS competent cells (MilliporeSigma) upon induction with 1 mM IPTG. After overnight growth, cells were harvested, suspended in a pH 8.0 lysis buffer containing 50 mM Tris, 6 M guanidine-HCl, 250 mM NaCl, 50 mM Imidazole, and 5 mM β-mercaptoethanol, bound to Ni-NTA resin, and washed with four-to-five column volumes. Protein was eluted using a linear gradient of 700 mM imidazole. Fractions containing pure FUS LCD were pooled and dialyzed overnight with a 20 mM CAPS pH 11.0 buffer. Stocks were concentrated to ~450 μM, flash frozen, and stored at −80 °C until use. FUS LCD concentration was determined via ε of 35,760 after diluting the stock into a pH 8 buffer (as protonation of FUS tyrosine residues at pH 11[68] can alter the absorption spectrum).

Full-length TDP-43 with C-terminal MBP (connected to TDP-43 through a linker containing TEV protease cleavage site) and His-tag was a gift from Nicolas Fawzi (Addgene plasmid #104480)[60]. Protein was expressed in Rosetta™ (DE3) pLysS competent cells (MilliporeSigma) upon induction with 1 mM IPTG. After 16 h growth at 15 °C, cells were harvested by centrifugation, suspended in lysis buffer (20 mM Tris, pH 8, 1 M NaCl, 10% glycerol, 10 mM imidazole, 1 mM PMSF, 1 mM DTT), bound to Ni-NTA resin, and washed with five column volumes. Protein was eluted without gradient using 500 mM imidazole. Fractions were immediately pooled and bound to amylose resin. Bound protein was washed with ten column volumes of amylose binding buffer (50 mM Tris, pH 7.4, 1 M NaCl, 10% glycerol, 1 mM PMSF, 1 mM DTT). Protein was eluted with the same buffer but additionally containing 10 mM D(+)-maltose. Fractions containing highly pure full-length TDP-43 were concentrated and further purified via gel filtration (buffer: 20 mM Tris, pH 8, 300 mM NaCl, 1 mM DTT) in order to separate higher-ordered TDP-43 aggregates from phase-separation competent TDP-43[60]. These latter fractions were were pooled and concentrated to 75 μM, flash frozen, and stored at −80 °C.

**Preparation of protein samples for liquid–liquid phase separation experiments.** In preparation for TDP-43 LCD LLPS experiments, lyophilized protein (both tagged or untagged) was dissolved in Milli-Q H$_2$O and passed through a 0.5 mL 100 kDa molecular weight cut-off (MWCO) Amicon Ultra centrifugal filter unit. Filtered protein concentration was calculated from the absorbance at 280 nm and the extinction coefficient (ε) of 17,990. Protein was then diluted into a buffer containing 20 mM potassium phosphate (pH 6.0) and any other reagents as needed for individual experiments. All turbidity measurements were performed within

5–10 min of sample preparation. All microscopy was performed within 30 min of sample preparation. To induce robust droplet formation in the absence of bis-ANS, protein was diluted to a final concentration of 20 μM in pH 8 buffer containing 20 mM potassium phosphate and 5% dextran.

In preparation for experiments with tau, protein was thawed on ice and diluted to reach a final concentration of 10 μM in a pH 7.4 buffer containing 10 mM HEPES, 10 mM NaCl, 1 mM DTT, and varying concentrations of bis-ANS. To induce robust droplet formation in the absence of bis-ANS, the same buffer with the addition of 10% dextran and 2 mM ethylenediaminetetraacetic acid (EDTA) was used.

For LLPS experiments with Ded1p, protein stored in a pH 8 buffer containing 50 mM Tris, 1 mM EDTA, 50% glycerol, 0.1% Triton-X, 2 mM DTT, and 300 mM NaCl was thawed on ice and subsequently diluted 25-fold to a final concentration of 1 μM in a helicase reaction buffer (40 mM Tris pH 8, 0.5 mM MgCl$_2$, 0.01% IGEPAL, 2 mM DTT) containing 150 mM NaCl and varying concentrations of bis-ANS. To induce robust droplet formation in the absence of bis-ANS, protein was diluted to a final concentration of 2 μM in helicase reaction buffer additionally containing 5% dextran and 30 mM NaCl.

For experiments with FUS, FUS LCD stocks were thawed on ice and diluted to 20 μM in a pH 8 buffer containing 20 mM potassium phosphate and varying concentrations of bis-ANS. For experiments with full-length FUS, MBP-FUS aliquots were thawed on ice and subsequently desalted into a pH 7.4 buffer containing 50 mM Tris, 2 mM DTT, 5% glycerol, and 150 mM NaCl using a 5 mL Zeba spin desalting column (7 K MWCO, Thermo Scientific). Concentration after desalting was measured by absorbance at 280 nm (ε = 139,470). Protein was diluted into a buffer to reach the final conditions of 25 mM Tris pH 7.4, 150 mM NaCl, 5% dextran, and 1 mM DTT with varying concentrations of bis-ANS.

For experiments with full-length TDP-43, the protocol for induction of LLPS was adopted from Wang et al.[60]. In short, frozen stocks of TDP-43 fused to MBP were thawed on ice and immediately diluted into a pH 7.0 buffer containing 20 mM HEPES, 1 mM DTT, 2 units TEV protease (NEB), a final NaCl concentration of 90–100 mM, and with or without 20 μM bis-ANS. Turbidity increase upon cleavage of MBP was monitored over time and bright field microscopy images were acquired between 30 and 60 min.

Small molecules used in LLPS experiments were obtained from the following sources: 4,4′-dianilino-1,1′-binaphthyl-5,5′-disulfonic acid (bis-ANS dipotassium salt; Invitrogen, B153), 8-anilinonaphthalene-1-sulfonic acid (ANS; Invitrogen, A47), 5-amino-1-naphthalenesulfonic acid (AmNS; Sigma, 70800), sodium 2-naphthalene sulfonate (SNS; TCI Chemicals, N0016), sodium benzene sulfonate (Sigma, 147281), c-di-AMP sodium salt (Sigma, SML1231), heparin disaccharide I-H sodium salt (Sigma, H8892), enoxaparin sodium (heparin; Sigma, E0180000), poly(A) potassium salt (Sigma, P9403), Dextran 70 (Sigma, 31390).

**Fluorescence and bright field microscopy.** All fluorescence images were acquired and analyzed using a Leica TCS SP8 confocal microscope with ×63 oil immersion lens and Leica Application Suite X (3.3.0.16799), unless otherwise stated. The excitation and emission maxima of bis-ANS in the presence of protein were measured by spectrofluorometer to be ~395 and ~500 nm, respectively. Therefore, generic fluorescence imaging (i.e., data acquisition that does not involve collecting consecutive images of the same sample or droplet) was performed by exciting bis-ANS with a 405 nm laser and recording emission between 500 and 545 nm via photomultiplier tube detector. Colocalization of bis-ANS and AF594-labeled TDP-43 LCD was performed using a sequential scanning (between lines) protocol to minimize cross-talk between fluorophores. Acquisition sequence was designed to first excite AF594-labeled TDP-43 LCD using a 594 nm laser, followed by excitation of bis-ANS. To minimize the problem of bis-ANS photobleaching[69] during experiments that required consecutive imaging of the same sample region (i.e., time-lapse capture of fusion events, FRAP), bis-ANS fluorescence image acquisition was performed using excitation wavelengths slightly above the bis-ANS excitation maximum (458 nm laser). Each FRAP experiment involved five pre-bleach frames, followed by one bleach frame, and recovery was monitored over 45 s (1.3 frame/s). Analysis of recovery involved correcting for photobleaching using reference fluorescence intensity from an unbleached droplet, in accordance with standard published protocols[70]. In short, intensity traces for experimental and reference droplets were normalized to the pre-bleach time points and the normalized intensity trace for the bleached droplet was divided by that of the reference droplet to correct for photobleaching. Traces were recorded either immediately or within 3 h after sample preparation. For Supplementary Fig. 9, a Keyence BZ-X710 fluorescence microscope with ×100 oil immersion lens and TexasRed filter cube (OP-87765) were used along with BZ-X Viewer (ver. 1.3.1.1) and BZ-X Analyzer (ver 1.3.1.1) software.

All bright field imaging was performed using a Keyence BZ-X710 fluorescence microscope with ×100/1.45 oil immersion lens or Leica TCS SP8 confocal microscope with ×63 oil immersion lens. Post-acquisition processing of images acquired using the BZ-X710 included a haze reduction filter.

**Turbidity, fluorescence intensity, and FRET measurements.** Turbidity (detected as absorbance at 600 nm) and fluorescence intensity measurements were carried out at 25 °C using a Tecan Spark multimode microplate reader with Te-Cool™ active temperature regulation. Fluorescence intensity measurements for FRET

experiments were carried out by excitation of sample at 280 nm and measuring emission at 356 nm (bandwidth: 20 nm) on the Tecan Spark. Representative fluorescence emission spectra for FRET experiments were acquired using a Horiba Scientific Jobin Yvon spectrofluorometer with an excitation wavelength of 280 nm.

**Electrophoretic mobility measurements**. Experiments were performed using a Malvern Zetasizer Nano ZS with a DTS1070 cell and analyzed using Malvern Zetasizer Software (ver. 7.13). Samples were prepared using 5 µM protein with varying concentrations of bis-ANS or poly(A). Measurements were acquired at 30 V with 20 runs per measurement. For each condition, measurements for three separately prepared samples were taken.

**Cell culture**. HCT-116 cells (ATCC CCL-247) were cultured in McCoy's 5A medium with L-glutamine and sodium bicarbonate (Sigma M9309), supplemented with penicillin and streptomycin (ThermoFisher 15140122), and 10% fetal bovine serum (ThermoFisher 26140079). Cells were incubated at 37 °C with 5% $CO_2$ and passaged every 3 days at a 1:12 ratio. Cell viability was routinely monitored via Trypan blue exclusion to ensure cell health was maintained throughout culturing. Mycoplasma PCR testing indicated that the cells were free of mycoplasma contamination.

**MTT assay**. Cell viability in the presence of bis-ANS was measured using the Cell Proliferation Kit 1 (MTT, 3-(4,5-dimethylthiazol-2-yl)-2,5-diphenyltetrazolium bromide) (Roche; cat. no. 11465007001) according to the manufacturer-suggested protocol. Briefly, $2 \times 10^4$ HCT-116 cells/well were seeded onto a 96-well, clear bottom, tissue culture-treated polystyrene plate (Corning Life Sciences; cat. no. 353072). Cells were incubated for 48 h at 37 °C, 5% $CO_2$ until each well was >90% confluent. Bis-ANS ranging from 0 to 1 mM was diluted into complete culture medium. Once cells reached the desired confluence, culture medium was removed and 100 µL bis-ANS containing media was added to cells for either 6 or 24 h. After bis-ANS treatment, 10 µL of MTT labeling reagent was added to each well and incubated at 37 °C for 4 h. After 4 h, 100 µL of solubilization solution was added to each well and incubated at 37 °C overnight. Absorbance was measured on a Tecan Microplate reader at 600 nm, using a 700 nm reference wavelength. To calculate the cell viability, corrected absorbance values (OD600−OD700) were divided by the control absorbance value (cells not treated with bis-ANS) then multiplied by 100 to obtain the percent viability. The above assay was repeated with two biological replicates (each with 2−3 technical replicates).

**Live-cell imaging**. Confluent HCT-116 cells were passaged onto 35 mm glass bottom culture dishes (Greiner bio one; cat. no. 627861), and allowed to grow until 70% confluence overnight. Cells were treated with bis-ANS or sodium arsenite dissolved in culture medium, and before imaging, cells were washed three times with Dulbecco's phosphate-buffered saline (Gibco; cat. no. 14190-144), then imaged in Fluorobrite™ Dulbecco's modified eagle medium (Gibco; cat. No. A18967-01) cell culture medium. In order to visualize the cell nuclei, the cell permeant Draq5 (Thermo Scientific; 62254) was added to the imaging media at 1:1000 ratio. Cells were imaged with a ×40 water immersion objective on the Leica TCS SP8 confocal microscope, which had a live-cell incubation chamber set up to maintain sample at 37 °C and 5% $CO_2$. Bis-ANS was excited as previously described, and Draq5 was excited using a 633 nm laser and emission was collected from 675 to 725 nm.

**Stress granule washout experiments**. In preparation for washout experiments, 200,000 cells were plated per well in a six-well plate containing glass coverslips (~18 mm diameter, round). Forty-eight hours later, cells were treated with 500 µM sodium arsenite for 30 min. Cells were then washed 3× with phosphate-buffered saline (PBS) and treated with media containing either PBS (control) or 250 µM bis-ANS for indicated times. Cells were then fixed for 15 min with 4% formaldehyde in PBS, washed 3× with PBS, blocked for 1 h at room temperature (0.3% Triton-X and 5% goat serum in PBS), and incubated overnight with 1:50 anti-G3BP (Sigma, WH0010146M1) at 4 °C. Cells were then washed 3× with PBS, incubated with 1:200 anti-mouse CF488 (Sigma, SAB4600237) for 2 h at room temperature, washed 3× with PBS, stained with DAPI, washed 3× with PBS, and mounted on slides with Fluoromount-G (Southern Biotech). Slides were imaged using a Leica DM6000 upright microscope. Twenty images per condition per replicate (≥200 cells) were analyzed for the percentage cells containing stress granules using the cell counter feature of ImageJ.

**Statistical analysis**. All data are represented as mean values ± standard deviation. Calculations were performed using Microsoft Excel (ver. 2013, 2017) and graphs generated using Origin (ver. 2017, 2019). Unless otherwise stated, $n$ indicates the number of replicates performed in an experiment.

**Chemical structures**. Chemical structures for naphthalene sulfonate derivatives and other compounds were created using ChemDraw Professional (ver. 17.0).

**Reporting summary**. Further information on research design is available in the Nature Research Reporting Summary linked to this article.

## Data availability

Data supporting the findings of this manuscript are available from the corresponding author upon reasonable request. A reporting summary for this Article is available as a Supplementary Information file. Publically available data shown in Supplementary Table 1 were acquired from PubChem (https://pubchem.ncbi.nlm.nih.gov/) or ChemSpider (http://www.chemspider.com/). Source data are provided with this paper.

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

## Acknowledgements

We thank Raza Haider for critical review and editing of this manuscript. We also thank Dr. Timothy Dumm and Hyperion Materials & Technologies Inc. for assistance with electrophoretic mobility measurements and Dr. Nicolas Fawzi for advice regarding purification of full-length TDP-43. This work was supported by NIH grants F30 AG059350 (W.M.B.), R01 NS103848 (W.K.S.), RF1 AG061797 (W.K.S.), R35 GM118088 (E.J.), T32 NS077888, T32 GM007250 and the Paul Berg and Harland Wood Graduate Fellowship (B.K.D.). Microscopy work was supported by NIH ORIP grants S10 OD024996 (Leica TCS SP8) and S10 RR021228 (Leica DM 6000).

## Author contributions

W.M.B. and W.K.S. designed the research and wrote the manuscript. W.M.B. performed experiments in vitro with TDP-43 LCD, Ded1p, and FUS. W.M.B. and B.K.D. performed electrophoretic mobility experiments and purified TDP-43 LCD. S.V. and B.K.D. performed experiments in cell culture model. S.B. purified tau protein. W.M.B. and S.B. performed experiments with tau. A.A.P. and S.V. purified and provided Ded1P protein. E.J. provided oversight and guidance with Ded1p and cell culture experiments. All authors contributed to the revision of manuscript draft.

## Competing interests

The authors declare no competing interests.
