## [Peer Review File · Nature Communications]

REVIEWER COMMENTS

Reviewer #1 (Remarks to the Author):

This is an interesting manuscript wherein the authors describe that small molecules, which are well-known binders of hydrophobic patches in proteins and their assemblies, modulates phase separation of TDP43 LCD. The study identifies that Bis-ANS and Congo Red can induce LLPS of TDP43 LCD at low concentrations while preventing the same at higher concentrations. Utilizing standard techniques such as solution turbidity and fluorescence microscopy, the authors characterized the effect of Bis-ANS induced reentrant LLPS of this RNP. Utilizing the intrinsic fluorescence properties of Bis-ANS, the authors showed that the small molecule co-localizes within TDP43 LCD droplets. To elucidate the mechanism of this observed effect, the authors studied several naphthalene sulfonate compounds and observed a correlation between their valence and hydrophobicity, and their ability to induce TDP43 LCD LLPS. Finally, the authors attempted to show that this is not specific to TDP43 LLPS but generic in nature by conducting experiments with Ded1p and Tau.

Small molecule mediated control of protein LLPS is an emerging direction that is not only of interest in the pharmaceutical industry but also from a fundamental mechanistic perspective. As such, the study is meritorious. However, the results are not surprising and the depth of analysis is poor. The idea of reentrant LLPS has been previously described in the literature (Refs 14 and 18) for RNP-RNA systems and has been postulated to be a generic phenomenon for systems undergoing LLPS via obligate heterotypic interactions by Pappu group (doi.org/10.1371/journal.pcbi.1007028). The later paper has not been cited in this manuscript. With respect to small molecules, ATP has been shown to produce a similar effect (Ref 41 in this manuscript) for an analogous protein FUS. Given these prior studies, the current observations are interesting but not surprising without an in depth mechanistic model. The following points might be worth considering to improve this study:

- a) How TDP43 LCD is interacting with RNA/heparin without any RNA binding domain? Can the authors comment and expand on that?
- b) Can Bis-ANS dissolve preformed homotypic TDP43 LCD condensates?
- c) Is the effect of Bis-ANS preserved for full-length TDP43 LLPS? This is an important question since Nick Fawzi's group has previously shown that the N-terminal of TDP43 also plays an important role in TDP43 LLPS. So, if a small molecule is desired as a modulator for TDP43 LLPS, it should be more valuable if the full-length protein is targeted.
- d) In the introduction section of the paper, the authors describe that the prevention of aberrant LLPS is desired rather than perturbing functional LLPS. Both TDP43 and FUS are known to undergo maturation into gel/fiber like condensates in vitro. Therefore, a natural question is whether Bis-ANS and analogous compounds can delay/prevent such a process? Prior studies from the Ferreon group showed that TMAO can facilitate condensation of TDP43 LCD but prevents its aggregation (<https://pubs.acs.org/doi/10.1021/acs.biochem.8b01051>). This is a highly relevant citation for the current study (has not been cited in this manuscript) and should be carefully considered. Therefore, the authors should show the effect of Bis-ANS in condensate gelation and fibrillation.
- e) The section describing TDP43 LCD LLPS with poly(A) RNA and Heparin is not well described. Does the TDP43 LCD is known to bind RNA/heparin? Is TDP43 LCD under investigation is a positively charged polyelectrolyte (As far as I understand, the answer is a NO based on the Fig S3)? If not properly addressed, it will be a big puzzle for the readers to understand how TDP43 LCD may undergo charge inversion and RNA/heparin dependent reentrant LLPS.
- f) TDP43 LCD LLPS with RNA and their modulation by salt are all very interesting pieces of data, but provide very little insight to the small-molecule dependent LLPS of TDP43 LCD.
- g) The molecular crowding experiments provide very little new information that could not be obtained from the other experiments reported in this paper.
- h) The discussion regarding ATP and Bis-ANS should be highlighted more, apart from just their effective concentrations required for modulating protein LLPS. I am also curious if ATP would behave similarly at physiological concentrations as Bis-ANS. The observed difference in the concentration

between these two compounds could stem from their differential solvation properties. Hence, a comparative measurement of K_d values will be required to shed light on these differences and provide more mechanistic insights.

i) Can the authors describe how Bis-ANS and related compounds can be utilized in vivo and in multi-component condensates as typically encountered in a cell? Would Bis-ANS have a similar effect for protein-RNA condensates instead of just protein only condensates?

Reviewer #2 (Remarks to the Author):

In their manuscript Babinchak et al. establish how small molecules can modulate the phase behaviour of proteins which are prone to liquid-liquid phase separation. The results of this work will likely improve our understanding of protein LLPS and may lead to new tools in cell biology. Small molecules have been known to influence peptides/proteins' propensity to undergo LLPS where some compounds enhance protein condensation while others dissolve protein condensates. Typically charge based mechanisms of condensation are characterised by a strong salt dependence. Other mechanisms of condensations rely on different chemical interactions and small molecules to interfere with these interactions have not been identified.

Here, it is convincingly demonstrated that small molecules can induce and inhibit protein LLPS in a concentration dependent manner, where condensation is not charge based alone. The underlying chemical design principles are identified and applied.

The present work provides an essential step to understand interaction specific protein LLPS.

Remarks and Questions:

1) The variety of low complexity domains and motifs which lead to protein LLPS or protein gelation is large. A better understanding of how peptide sequence determines phase behaviour will be critical for the field to make progress. In this context it would be great if the authors would discuss which peptide sequence(s) would be more or less sensitive to their small molecule phase modulation. Are there specific motives which respond particularly strong? What are the identified mechanisms for the proteins that are being studied in the manuscript? It would be great if, by addressing such questions, the authors would highlight the biochemical side of protein LLPS. In addition to some readers it may be useful to also include the relevant amino acid sequences in the text.

2) There are multiple questions that arise from the presented results. It would be very useful to the field of phase separation if the authors would more clearly discuss these emerging questions. Would it be important to perform molecular dynamics/quantum mechanics simulations in order to identify even more components? Could the design principles for small molecules help to understand protein-protein interactions towards multi-protein LLPS?

3) In the cell biological context it would be desirable to know if the identified compounds can actually enter cells and if so which range of concentration would be bearable for cells? While it may not be possible for the authors to test this for complex cell biology questions, I do not see why it should not be possible to try these compounds in a simple yeast cell culture. If the compounds enter the cell they would readily colocalize to MLOs and could be observed under the microscope. It seems like a very easy thing to do and the manuscript would greatly benefit if such an experiment was described. If such or similar experiments are not possible, the authors should discuss their expected difficulties and outcomes in the paper to facilitate others to pursue this task.

I enjoyed reading this manuscript.

Kind regards,
Louis Reese

Reviewer #3 (Remarks to the Author):

The authors use a large number of compounds (focused on bis-ANS) and additionally co-solutes and co-factors of different kinds to systematically test their effects on TDP-43 LCD (also microtubule-binding protein tau and FUS) liquid-liquid phase separation. Their studies showed that bis-ANS promotes protein droplet formation, but can also inhibit LLPS that is associated with pathological protein aggregation. As the authors claim, I agree that their studies give an insight into the general chemical principles or design criteria that could be utilized for identification of other LLPS-modulating compounds.

I enjoyed reading the work, however, I very much missed studies that are performed on the cellular level to show the relevance for biological systems. This is important as a lot of compounds are already known (or are expected) to affect LLPS. Compound development is nowadays commonly conducted on the cellular level especially for LLPS process which are well amenable by imaging. I think these experiments would be rather simple for the systems the authors choose to work with.

Another point is that the choice of experimental techniques used for the individual compounds tested is not clear to me and seems to be a bit random. Why is only a single FRAP experiment reported, (Fig 1g), that claims a liquid character of the droplet. How were the data analysed, what are the controls, why have such studies not been conducted for the other compounds (building blocks of bis-ANS, Congo red, ...)? Another example along the same line would be the FRET experiments.

Point-by point response to reviewers' comments

We wish to thank all three reviewers for their constructive comments and suggestions. We believe that addressing these comments have substantially improved the manuscript.

The changes we made are marked in red in the revised manuscript.

Reviewer #1:

This is an interesting manuscript wherein the authors describe that small molecules, which are well-known binders of hydrophobic patches in proteins and their assemblies, modulates phase separation of TDP43 LCD. The study identifies that Bis-ANS and Congo Red can induce LLPS of TDP43 LCD at low concentrations while preventing the same at higher concentrations. Utilizing standard techniques such as solution turbidity and fluorescence microscopy, the authors characterized the effect of Bis-ANS induced reentrant LLPS of this RNP. Utilizing the intrinsic fluorescence properties of Bis-ANS, the authors showed that the small molecule co-localizes within TDP43 LCD droplets. To elucidate the mechanism of this observed effect, the authors studied several naphthalene sulfonate compounds and observed a correlation between their valence and hydrophobicity, and their ability to induce TDP43 LCD LLPS. Finally, the authors attempted to show that this is not specific to TDP43 LLPS but generic in nature by conducting experiments with Ded1p and Tau.

Small molecule mediated control of protein LLPS is an emerging direction that is not only of interest in the pharmacological industry but also from a fundamental mechanistic perspective. As such, the study is meritorious. However, the results are not surprising and the depth of analysis is poor. The idea of reentrant LLPS has been previously described in the literature (Refs 14 and 18) for RNP-RNA systems and has been postulated to be a generic phenomenon for systems undergoing LLPS via obligate heterotypic interactions by Pappu group (doi.org/10.1371/journal.pcbi.1007028). The later paper has not been cited in this manuscript. With respect to small molecules, ATP has been shown to produce a similar effect (Ref 41 in this manuscript) for an analogous protein FUS. Given these prior studies, the current observations are interesting but not surprising without an in depth mechanistic model. The following points might be worth considering to improve this study:

We wish to thank the reviewer for carefully reading our manuscript and appreciate his/her overall positive assessment of the manuscript and many detailed and thoughtful comments and recommendations for additional experiments and points for discussion. We are somewhat puzzled by one of the comments that the results are “not surprising”, as—to the best of our knowledge—this is the first report that small molecules may act as biphasic modulators of protein LLPS.

With regard to an “in depth mechanistic model”, we have performed additional experiments as suggested by the reviewer (see our response to specific points below) and believe that these new data have not only further strengthen our conclusions but also substantially improved mechanistic insight, especially with regard to the nature of interactions responsible for the modulatory action of small molecules on protein LLPS.

Furthermore, we have now made it clear in the Introduction that reentrant LLPS has been postulated to be a generic phenomenon for systems undergoing LLPS via heterotypic interactions (p. 3). We have also cited the Pappu group paper in question, as it does bear conceptual relevance to our LLPS systems involving heparin and RNA. However, it is presently

unclear how such a model would apply to reentrant phase transitions with small molecule-mediated LLPS systems, in which these compounds do not have a so-called “stickers-and-spacers architecture.” With regard to the effect of ATP on LLPS, we address this issue below in our response to point (h).

The following points might be worth considering to improve this study:

a) How TDP43 LCD is interacting with RNA/heparin without any RNA binding domain? Can the authors comment and expand on that?

This aspect of our study indeed may require better explanation. RNA is known to interact electrostatically with many basic proteins that do not have specific RNA binding motifs. Our salt-dependence data shown in Fig. 3 strongly indicate that the interaction between TDP-43 LCD and RNA/heparin is also charge-based. Because RNA/heparin can only contribute negative charges (either from negatively charged phosphate, sulfate, or π -groups), the specific moieties contributed by TDP-43 for this interaction must therefore be positively charged side chains. We have added this clarification to the discussion section on p. 19.

b) Can Bis-ANS dissolve preformed homotypic TDP43 LCD condensates?

This is a great suggestion. We have performed the recommended experiment, finding that bis-ANS can indeed disrupt homotypic TDP43 LCD condensates formed in the presence of a crowding agent. These data are shown in Supplementary Figure 8a and described in the main text on p. 15.

c) Is the effect of Bis-ANS preserved for full-length TDP43 LLPS? This is an important question since Nick Fawzi's group has previously shown that the N-terminal of TDP43 also plays an important role in TDP43 LLPS. So, if a small molecule is desired as a modulator for TDP43 LLPS, it should be more valuable if the full-length protein is targeted.

Again, a very good suggestion. We adopted Dr. Fawzi's protocol for inducing LLPS of the full-length TDP-43, finding that bis-ANS can indeed disrupt the ability of full-length TDP-43 to undergo LLPS (that otherwise occurs upon cleavage of the maltose binding protein tag fused to TDP-43). Consistent with the Fawzi study, we also observed that full-length TDP-43 LLPS was followed by relatively fast aggregation of the protein. Importantly, this aggregation event was also prevented by bis-ANS. These new data are shown in Supplementary Figure 8e and described on p.15.

d) In the introduction section of the paper, the authors describe that the prevention of aberrant LLPS is desired rather than perturbing functional LLPS. Both TDP43 and FUS are known to undergo maturation into gel/fiber like condensates in vitro. Therefore, a natural question is whether Bis-ANS and analogous compounds can delay/prevent such a process? Prior studies from the Ferreon group showed that TMAO can facilitate condensation of TDP43 LCD but prevents its aggregation <https://pubs.acs.org/doi/10.1021/acs.biochem.8b01051>). This is a highly relevant citation for the current study (has not been cited in this manuscript) and should be carefully considered. Therefore, the authors should show the effect of Bis-ANS in condensate gelation and fibrillation.

The main focus of this study was on the effect of bis-ANS and related molecules on LLPS *per se*, and not that much on the consequences of the condensation with regard to protein gelation. Nevertheless, the reviewer raises a valid question. To at least partially address this question, we

now include additional FRAP studies. These data indicate that TDP-43 LCD droplets in the presence of bis-ANS retain highly dynamic nature even 3 hours after formation. This contrasts with the behavior of droplets formed without bis-ANS, which were previously shown by us and others to lose dynamicity (as assessed by FRAP) within ~45 min. These new data are shown in Fig. 1g and described on p. 6 of the revised manuscript. Furthermore, our new experiments with full-length TDP-43, where cleavage of MBP fused to TDP-43 in the absence of bis-ANS induces not only LLPS but subsequent aggregation speak to the ability for bis-ANS to disrupt aggregation as well. Indeed, in sharp contrast to observations in the absence of bis-ANS, no time-dependent increase in turbidity is observed in the presence of bis-ANS (see Supplementary Figure 8e and p. 15).

Ideally, one would want to further confirm these observations using a direct fibrillation assay such as Thioflavin-S or T fluorescence (employed by the Ferreon group and our lab (Babinchak et al. *JBC* 2019) previously). However, bis-ANS (which fluoresces at the same wavelengths as ThT dyes) strongly interferes with these assays. We hope the reviewer will appreciate this point, especially since the main focus of our present study is on LLPS per se, not on protein aggregation.

With regard to the study of the Ferreon group, we indeed failed to cite this paper. However, we would like to point out that the effect of TMAO only occurs at molar concentrations, in sharp contrast to small molecules such as bis-ANS which act in μM concentrations. We now clarify this point (and cite the Ferreon paper) on p. 4.

e) The section describing TDP43 LCD LLPS with poly(A) RNA and Heparin is not well described. Does the TDP43 LCD is known to bind RNA/heparin? Is TDP43 LCD under investigation is a positively charged polyelectrolyte (As far as I understand, the answer is a NO based on the Fig S3)? If not properly addressed, it will be a big puzzle for the readers to understand how TDP43 LCD may undergo charge inversion and RNA/heparin dependent reentrant LLPS.

We appreciate this comment, as this issue indeed requires better clarification. Given that the net positive charge of TDP-43 LCD is relatively small, it is indeed somewhat puzzling that negatively charged polymers such as poly(A) and heparin (as found in this study) or RNA (as determined previously) can induce LLPS. However, in our opinion, there is no other explanation of this effect than attractive electrostatic interactions between TDP-43 LCD and these polyanionic polymers. We have added additional clarification of this issue on p. 7 and 19.

f) TDP43 LCD LLPS with RNA and their modulation by salt are all very interesting pieces of data, but provide very little insight to the small-molecule dependent LLPS of TDP43 LCD.

The reviewer is right that data in the presence of poly(A) are indeed somewhat unrelated to our main focus on LLPS modulation by small molecules. However, we feel that these data are very helpful for the average reader, as they provide a “baseline” illustrating how LLPS driven by charge-based interactions can be manipulated by NaCl. In contrast to the TDP-43 LCD/poly(A) system, salt has essentially no effect on LLPS of TDP-43 LCD in the presence of bis-ANS, indicating that the latter system is fundamentally different with regard to the type of interactions involved.

g) The molecular crowding experiments provide very little new information that could not be obtained from the other experiments reported in this paper.

We agree with the reviewer that, from a purely mechanistic perspective, most of the information can be inferred from experiments in the absence of molecular crowders. Nevertheless, we strongly feel that data in the presence of crowding agents are an important part of our study, as they allow us to demonstrate that bis-ANS can abrogate condensation of many proteins under the conditions that favor homotypic LLPS. Furthermore, the conditions of these experiments mimic the crowded intracellular environment. We have revised the paragraph on p. 15 to better clarify this point.

h) The discussion regarding ATP and Bis-ANS should be highlighted more, apart from just their effective concentrations required for modulating protein LLPS. I am also curious if ATP would behave similarly at physiological concentrations as Bis-ANS. The observed difference in the concentration between these two compounds could stem from their differential solvation properties. Hence, a comparative measurement of Kd values will be required to shed light on these differences and provide more mechanistic insights.

Careful examination of the literature dealing with the effect of ATP on protein LLPS (refs. 45-48) shows that only one of these studies (Kang et al, BBRC 2018) claimed that ATP can induce LLPS (of FUS). However, limited data presented in this paper does not actually show any *de novo* induction of LLPS by ATP. Rather, ATP is shown only to “enhance” LLPS when droplets are already present for the full-length FUS. In order to more clearly delineate the LLPS-inducing capabilities of bis-ANS as compared to ATP, we performed additional experiments to determine whether ATP could induce LLPS for the TDP-43 LCD at higher concentrations. As shown in Supplementary Figure 6f, no such effect was observed even at ATP concentration as high as 10 mM. This strongly suggest that the action of bis-ANS (which exerts such an effect for both TDP-43 and FUS LCDs as well as four full-length proteins) is quite unique, clearly different from that described in previous studies for ATP. We have further clarified this on p. 12 of the revised manuscript. Given that ATP has absolutely no LLPS-inducing effect in our system, it is not clear to us what one could learn from comparative measurements of Kd values for ATP and bis-ANS. We hope the reviewer will understand and appreciate this point.

i) Can the authors describe how Bis-ANS and related compounds can be utilized in vivo and in multi-component condensates as typically encountered in a cell? Would Bis-ANS have a similar effect for protein-RNA condensates instead of just protein only condensates?

To address this question, we have performed additional experiments to test the effect of bis-ANS on LLPS of TDP-43 in the presence of poly(A). These new data clearly indicate that bis-ANS can also disrupt the formation of these complex, physiologically more relevant condensates (see Supplementary Figure 4c and p. 10). Further, we now additionally show that bis-ANS can be utilized in cellular systems to modulate stress granules (see Fig. 7 and Supplementary Fig. 9). These additional data are briefly described below in our response to Reviewer #2 and, in greater detail, on p. 16-17 of the main text.

Reviewer #2:

In their manuscript Babinchak et al. establish how small molecules can modulate the phase behaviour of proteins which are prone to liquid-liquid phase separation. The results of this work will likely improve our understanding of protein LLPS and may lead to new tools in cell biology. Small molecules have been known to influence peptides/proteins' propensity to undergo LLPS where some compounds enhance protein condensation while others dissolve protein condensates. Typically charge based mechanisms of condensation are characterised by a

strong salt dependence. Other mechanisms of condensations rely on different chemical interactions and small molecules to interfere with these interactions have not been identified. Here, it is convincingly demonstrated that small molecules can induce and inhibit protein LLPS in a concentration dependent manner, where condensation is not charge based alone. The underlying chemical design principles are identified and applied. The present work provides an essential step to understand interaction specific protein LLPS.

We wish to thank the reviewer for careful reading of our manuscript and appreciate his/her overall positive assessment of our work as summarized in the concluding statement “The underlying chemical design principles are identified and applied. The present work provides an essential step to understand interaction specific protein LLPS”.

1) The variety of low complexity domains and motifs which lead to protein LLPS or protein gelation is large. A better understanding of how peptide sequence determines phase behaviour will be critical for the field to make progress. In this context it would be great if the authors would discuss which peptide sequence(s) would be more or less sensitive to their small molecule phase modulation. Are there specific motives which respond particularly strong? What are the identified mechanisms for the proteins that are being studied in the manuscript? It would be great if, by addressing such questions, the authors would highlight the biochemical side of protein LLPS. In addition to some readers it may be useful to also include the relevant amino acid sequences in the text.

This is a very good suggestion. Given that the effect of bis-ANS on LLPS is observed for a number of proteins (TDP-43, FUS, Ded1p tau) that do not contain any consensus sequence motifs, it appears that this effect is relatively nonspecific with regard to the amino acid sequence (even though this issue may deserve further examination using a larger number of proteins). We have added some additional discussion of these aspects in the discussion section on p. 20-21. Additionally, our studies indicate that sequence specificity is not a very important factor in the modulation of protein LLPS by bis-ANS. We therefore feel that including amino acid sequences of all proteins used may be redundant, especially since these sequences can be readily found in publically-available databases.

2) There are multiple questions that arise from the presented results. It would be very useful to the field of phase separation if the authors would more clearly discuss these emerging questions. Would it be important to perform molecular dynamics/quantum mechanics simulations in order to identify even more components? Could the design principles for small molecules help to understand protein-protein interactions towards multi-protein LLPS?

If we understand this point correctly, the reviewer asks us to discuss future directions in the emerging efforts to develop drugs that modulate LLPS. Given that this field is still in its infancy, this discussion would need to be somewhat speculative and we feel a little reluctant to venture in this direction, especially in this manuscript that focuses on experimental data. We hope the reviewer will understand and appreciate our position in this regard.

Regarding the role of molecular dynamics/quantum mechanics simulations, in our understanding computational approaches to drug development work best in cases where there are well-defined protein binding sites for small molecules. Given that binding of bis-ANS and related molecules to proteins is relatively nonspecific, we feel that these approaches might not be readily applicable to this system (even though none of the authors has expertise in this area).

The question regarding multi-protein LLPS is a good one. This is a complex issue that requires many additional studies (that are the area of future directions of our research). We hope the reviewer will appreciate that addressing this issue is somewhat beyond the scope of the present manuscript. We now briefly comment on this issue at the end of the Discussion section (p. 21).

3) In the cell biological context it would be desirable to know if the identified compounds can actually enter cells and if so which range of concentration would be bearable for cells? While it may not be possible for the authors to test this for complex cell biology questions, I do not see why it should not be possible to try these compounds in a simple yeast cell culture. If the compounds enter the cell they would readily colocalize to MLOs and could be observed under the microscope. It seems like a very easy thing to do and the manuscript would greatly benefit if such an experiment was described. If such or similar experiments are not possible, the authors should discuss their expected difficulties and outcomes in the paper to facilitate others to pursue this task.

This is an important question and we have now addressed this aspect in a new Fig. 7 and Supplementary Fig. 9 (and described on p. 16-17 of the main text). In summary, we show that bis-ANS not only can readily enter the cell and is relatively non-toxic through 400 μ M concentrations, but also that the addition of bis-ANS can modulate the dissolution of stress granules formed by the addition of sodium arsenite. We felt that a mammalian cell (HCT-116) model would be most appropriate for these studies and we hope that the reviewer will find these results sufficient in lieu of a yeast cell culture model. We have additionally included a new paragraph that discusses these results (p. 21, final paragraph of Discussion).

Reviewer #3:

The authors use a large number of compounds (focused on bis-ANS) and additionally cosolutes and co-factors of different kinds to systematically test their effects on TDP-43 LCD (also microtubule-binding protein tau and FUS) liquid-liquid phase separation. Their studies showed that bis-ANS promotes protein droplet formation, but can also inhibit LLPS that is associated with pathological protein aggregation. As the authors claim, I agree that their studies give an insight into the general chemical principles or design criteria that could be utilized for identification of other LLPS-modulating compounds.

We appreciate reviewer's positive general comments about our manuscript and the overall assessment that this study "gives an insight into the general chemical principles or design criteria that could be utilized for identification of other LLPS-modulating compounds".

I enjoyed reading the work, however, I very much missed studies that are performed on the cellular level to show the relevance for biological systems. This is important as a lot of compounds are already known (or are expected) to affect LLPS. Compound development is nowadays commonly conducted on the cellular level especially for LLPS process which are well amenable by imaging. I think these experiments would be rather simple for the systems the authors choose to work with.

We agree with the reviewer that the relevance of these studies to cellular systems is an important factor. Even though the main focus of our manuscript still remains of studies *in vitro*, we have now additionally included data depicted in Fig. 7 and Supplementary Fig. 9 demonstrating that bis-ANS can indeed modulate LLPS within a cellular setting. To this end, we have utilized a mammalian cell model in which the addition and later removal of sodium arsenite results in stress granule formation and dissolution, respectively. Importantly, the addition of bis-

ANS delays the dissolution of these granules, indicating that the small molecule has relevance for biological systems. These results are described on p. 16-17 (see also our response to comment 3 of reviewer 2).

Another point is that the choice of experimental techniques used for the individual compounds tested is not clear to me and seems to be a bit random. Why is only a single FRAP experiment reported, (Fig 1g), that claims a liquid character of the droplet. How were the data analysed, what are the controls, why have such studies not been conducted for the other compounds (building blocks of bis-ANS, Congo red, ...)? Another example along the same line would be the FRET experiments.

With regard to the question why no FRAP (or FRET) experiment were performed for all compounds tested, it appears that we might have failed to clarify well enough that most of these additional compounds (building blocks of bis-ANS) have been used solely to determine which of the chemical features of bis-ANS are important for its high activity as a modulator of protein LLPS. Most of these compounds have very little activity in this regard, acting only at concentrations at least one order (most of them two orders) of magnitude higher compared to bis-ANS. Thus, given this low activity, we feel that testing the effect of these compounds on droplet dynamicity (i.e., FRAP experiments) wouldn't add much to our study.

Second, the presence of intrinsic fluorescence and rapid photobleaching of some these compounds can greatly obfuscate FRAP experiments. While the fluorescence of bis-ANS does photobleach at typical laser intensities (as discussed in our methods section), we could combat this by using very low laser intensities. However, this problem is much more significant for Congo red (a highly active molecule that would be worth testing with regard to droplet dynamicity), precluding FRAP experiments based on fluorescence of this molecule. We considered concurrently utilizing an AlexaFluor-labeled TDP-43 construct to facilitate such experiments; however, we found that the excitation spectrum of Congo red essentially encompasses the majority of the visible spectrum (and therefore most commonly-used fluorophores for FRAP studies). Thus, excitation of any fluorescent label would still result in photobleaching of Congo red – an effect that could not be parsed out during FRAP experiments. Given these technical difficulties, we assessed liquid-like character of droplets formed in the presence of Congo red by an alternative method, i.e., following fusion events (Fig. 5d), which are foundational observations in most studies of protein LLPS. For completeness, we now include fusion data not only for droplets formed in the presence of bis-ANS (Fig. 1f), but also in the presence of ANS (Supplementary Fig 6a), and SNS (Supplementary Fig. 6h). These new data are described on p. 11 (ANS) and 13 (SNS).

Regarding the question “*How were the data analysed, what are the controls?*”, we assume the reviewer is referring specifically to FRAP experiments. Analysis of these data followed a standard protocol frequently used in the LLPS field. We have now provided additional details in this regard in the Experimental Procedures section (p. 26).

REVIEWERS' COMMENTS

Reviewer #1 (Remarks to the Author):

In the revised manuscript by Babinchak, the authors have addressed several of the original concerns raised by this reviewer and added new experimental data and/or presented clearer arguments. However, I still have one concern at this stage. My previous question regarding "can Bis-ANS dissolve preformed TDP43 droplets?" is not clearly answered and needs further clarifications:

(a) Point b in the response letter: It is not clear whether Bis-ANS can dissolve preformed TDP43 droplets. The authors write "The addition of bis-ANS at higher concentrations completely abrogated the ability for the full-length TDP-43 to form droplets". This is not what I asked in the original report. The new data shown in Fig S8 appears to me that Bis-ANS prevents TDP43 droplet formation in presence of dextran, but can it REVERSE the TDP43 LLPS by dissolving this droplet? Time lapse imaging of TDP43 droplets after addition of Bis-ANS would allow answering this question. If it does not dissolve, then state it clearly in the manuscript.

Other than this, everything looks good. Thank you!

Reviewer #2 (Remarks to the Author):

In their revised manuscript and responses the authors address the reviewers comments and questions. I am satisfied with how my comments have been addresses. In particular, new data is presented which shows a cell culture experiment. It is shown that the identified compound can enter human cells and influence cellular phenomena, such as artificially induced stress granules. These additional experiments complement the initial findings very well and provide a complete picture regarding the usefulness of the authors' discovery.

Louis Reese

Reviewer #3 (Remarks to the Author):

The authors improved the manuscript by additional experiments and further clarifications. I recommend publication in its present form.

Point-by point response to reviewers' comments

We wish to thank all three reviewers for reviewing the manuscript and for their constructive comments and suggestions.

The revision we made in response to comments of reviewer #1 is marked in red on p. 15 of the revised manuscript.

Reviewer #1:

In the revised manuscript by Babinchak, the authors have addressed several of the original concerns raised by this reviewer and added new experimental data and/or presented clearer arguments. However, I still have one concern at this stage. My previous question regarding "can Bis-ANS dissolve preformed TDP43 droplets?" is not clearly answered and needs further clarifications:

(a) Point b in the response letter: It is not clear whether Bis-ANS can dissolve preformed TDP43 droplets. The authors write "The addition of bis-ANS at higher concentrations completely abrogated the ability for the full-length TDP-43 to form droplets". This is not what I asked in the original report. The new data shown in Fig S8 appears to me that Bis-ANS prevents TDP43 droplet formation in presence of dextran, but can it REVERSE the TDP43 LLPS by dissolving this droplet? Time lapse imaging of TDP43 droplets after addition of Bis-ANS would allow answering this question. If it does not dissolve, then state it clearly in the manuscript.

Other than this, everything looks good. Thank you!

As requested by the reviewer, we now provide a direct evidence that bis-ANS not only abrogates droplet formation but it can also reverse the TDP-43 LCD LLPS by dissolving preformed droplets. This evidence is provided in Supplementary Figure 9 and mentioned on p. 15 of the revised manuscript.

Reviewer #2:

In their revised manuscript and responses the authors address the reviewers comments and questions. I am satisfied with how my comments have been addresses. In particular, new data is presented which shows a cell culture experiment. It is shown that the identified compound can enter human cells and influence cellular phenomena, such as artificially induced stress granules. These additional experiments complement the initial findings very well and provide a complete picture regarding the usefulness of the authors' discovery.

Louis Reese

This reviewer is satisfied with our revisions.

Reviewer #3:

The authors improved the manuscript by additional experiments and further clarifications. I recommend publication in its present form.

This reviewer is satisfied with our revisions.